# REX-RAG: Reasoning Exploration with Policy Correction in Retrieval-Augmented Generation

## Abstract

Reinforcement learning (RL) is emerging as a powerful paradigm for enabling large language models (LLMs) to perform complex reasoning tasks. Recent advances indicate that integrating RL with retrieval-augmented generation (RAG) allows LLMs to dynamically incorporate external knowledge, leading to more informed and robust decision making. However, we identify a critical challenge during policy-driven trajectory sampling: LLMs are frequently trapped in unproductive reasoning paths, which we refer to as "dead ends", committing to overconfident yet incorrect conclusions. This severely hampers exploration and undermines effective policy optimization. To address this challenge, we propose **REX-RAG** (Reasoning Exploration with Policy Correction in Retrieval-Augmented Generation), a novel framework that explores alternative reasoning paths while maintaining rigorous policy learning through principled distributional corrections. Our approach introduces two symbiotic innovations: (1) Mixed Sampling Strategy, which combines a novel probe sampling method with exploratory prompts to escape dead ends; and (2) Policy Correction Mechanism, which is essential for correcting the distributional shifts introduced by exploration. REX-RAG demonstrates that effective exploration is only viable when paired with such a rigorous correction. We evaluate it on seven question-answering benchmarks, and the experimental results show that REX-RAG achieves average performance gains of 5.1% on Qwen2.5-3B and 3.6% on Qwen2.5-7B over strong baselines, demonstrating competitive results across multiple datasets. Anonymous repository is provided on `https://anonymous.4open.science/r/REX-RAG`.

## 1 Introduction

Recent advances have shown that reinforcement learning (RL) offers a promising avenue for training large language models (LLMs) to perform complex reasoning tasks (Ouyang et al., 2022; Chen et al., 2025b). By integrating multi-step reasoning with retrieval-augmented generation (RAG), RL-trained LLMs can dynamically leverage external knowledge sources—essentially allowing them to "think while searching" (Chen et al., 2025a; Jin et al., 2025b). This paradigm holds particular promise for multi-hop question answering, where models must iteratively gather and synthesize evidence across multiple queries to arrive at well-founded conclusions (Jin et al., 2025a).

Despite this potential, we observe a critical challenge that substantially hinders policy optimization in such settings. During RL training, LLMs frequently become trapped in what we term "*dead ends*", which is defined as a state in the reasoning process where all sampled trajectories consistently fail to reach a correct final answer. This phenomenon often stems from premature or overconfident conclusions drawn despite insufficient supporting information, effectively terminating exploration along potentially fruitful reasoning paths (Yue et al., 2025; Wen et al., 2025; Liu et al., 2025).

Addressing this challenge requires mechanisms that can proactively explore alternative reasoning paths when initial trajectories prove unproductive. A straightforward solution is *self-reflection* (Guo et al., 2025; Jin et al., 2025b), which attempts to revise failed reasoning chains to generate alternative ones. However, we observe that these revised trajectories are often merely slight perturbations of the original paths, offering limited novelty and insufficient deviation to meaningfully explore alternative

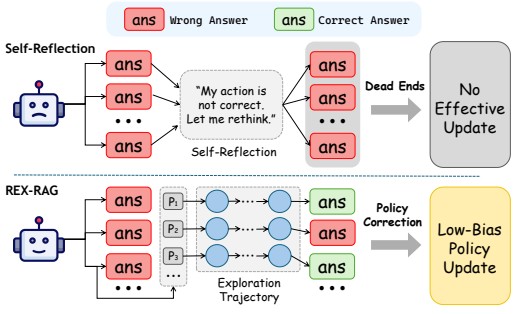
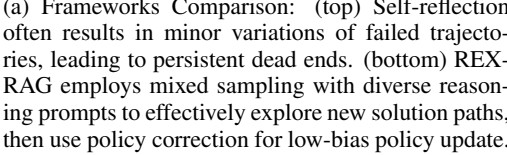
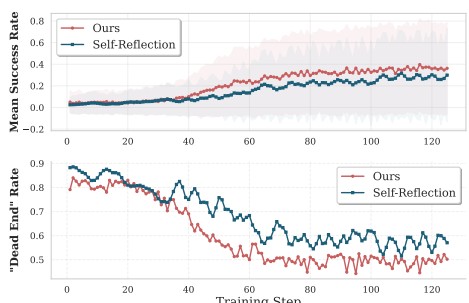

(a) Frameworks Comparison: (top) Self-reflection often results in minor variations of failed trajectories, leading to persistent dead ends. (bottom) REX-RAG employs mixed sampling with diverse reasoning prompts to effectively explore new solution paths, then use policy correction for low-bias policy update.

(b) Training Dynamics: REX-RAG (red) consistently achieves a higher success rate and a lower dead-end rate compared to the self-reflection baseline (blue) throughout training. The shaded areas correspond to the variance.

Figure 1: REX-RAG addresses the challenge of "dead ends" in RL-based RAG. Subfigure (a) illustrates how self-reflection fails to escape incorrect reasoning paths, while REX-RAG's guided exploration opens up new possibilities. Subfigure (b) provides empirical evidence, showing REX-RAG's superior performance in success rate and its effectiveness in reducing dead ends during training.

solutions. Consequently, it struggles to escapee from dead-end paths, as illustrated in Fig. 1(a). In our experiments with Qwen2.5-3B model on multiple datasets, self-reflection consistently results in a high incidence of "dead ends". This phenomenon surpasses 85% in the early phases (nearly first 50 epochs) of RL training and significantly impedes effective policy learning, as shown in Fig. 1(b).

On the other hand, more aggressively enforcing exploration, such as introducing additional agents (Xiong et al., 2025; Nguyen et al., 2025), makes end-to-end optimization challenging due to the complexity of jointly training multiple components. This challenge underscores the need for principled strategies that can foster sufficiently diverse and informative exploration while ensuring stable and unbiased policy optimization without compromising the end-to-end learning paradigm (Feng et al., 2025). This creates a fundamental exploration-optimization dilemma.

To address this challenge, we propose **REX-RAG** (**R**easoning **EX**ploration with Policy Correction in Retrieval-Augmented Generation), a novel framework that explores alternative reasoning paths while maintaining rigorous policy learning through principled distributional corrections. Our framework incorporates an exploratory probe policy that collaborates with the standard policy to escape from the "dead ends", as shown in Fig. 1 (a).

For exploration, REX-RAG introduces *Mixed Sampling Strategy*. Unlike self-reflection methods that result in minor variations of failed path, this strategy is designed to induce diverse reasoning trajectories. Specifically, when the policy encounters "dead end", it surgically injects a curated reasoning prompt, fundamentally altering the generation context. This forces the model to break from its failing logic and explore new solution paths, rather than merely re-attempting similarly.

Such an exploration strategy is only viable if its impact on policy optimization can be rigorously managed. This is achieved by *Policy Correction Mechanism*, which makes exploration stable and trainable. This mechanism unifies two distinct trajectories from the origin policy and the probe policy under a single, low-bias optimization objective. By leveraging importance sampling to precisely re-weight the contributions of each component in the trajectory, it corrects for the inherent distribution shift introduced by exploration (Yan et al., 2025; Tan et al., 2025).

Extensive experiments on multi-hop question answering benchmarks demonstrate that REX-RAG significantly outperforms existing methods, achieving substantial improvements in both answer accuracy and reasoning quality. On average, it outperforms strong baselines by 5.1% on Qwen2.5-3B and 3.6% on Qwen2.5-7B. Furthermore, as shown in Fig. 1(b), our analysis reveals that the framework successfully escapes dead ends while maintaining stable policy learning, with consistently higher success rates and lower dead end rates compared to self-reflection approaches, validating the effectiveness of our principled exploration strategy.

The main contribution can be concluded that: **(1)** We identify and formalize the "*dead end*" problem in RL-based RAG training, demonstrating its significant impact on policy optimization, posing a substantial obstacle to effective policy learning. **(2)** We propose **REX-RAG**, whose innovation lies in a symbiotic design that resolves the exploration-optimization dilemma in RL-based RAG. *Policy Correction Mechanism* underpins the principled exploration of *Mixed Sampling Strategy* by correcting the distributional shifts inherent, providing a stable, end-to-end solution that harmonizes these competing objectives. **(3)** We achieve substantial improvements over strong baselines (5.1% on Qwen2.5-3B and 3.6% on Qwen2.5-7B) on multiple open-domain QA benchmarks.

## 2 RELATED WORK

**Retrieval-Augmented Generation.** RAG (Lewis et al., 2020) has fundamentally transformed how language models access and utilize external knowledge. The RAG framework combines search engines with LLMs, enabling them to ground responses in retrieved documents (Arslan et al., 2024). This paradigm has proven particularly effective for knowledge-intensive tasks where parametric knowledge alone is insufficient (Mallen et al., 2023). For multi-hop reasoning tasks, several approaches have emerged (Asai et al., 2024; Gao et al., 2025), for example, IRCoT (Trivedi et al., 2023) interleaves retrieval with chain-of-thought reasoning, allowing models to iteratively gather evidence across multiple steps. These pioneering RAG methods have laid a strong foundation for subsequent RL-based approaches, which deeply integrate the retrieval and reasoning processes.

**Reinforcement Learning with Verifiable Rewards (RLVR).** The integration of RLVR and RAG has opened new avenues for training LLMs to perform complex reasoning tasks, and yielded impressive results (Zheng et al., 2025; Mei et al., 2025; Qian & Liu, 2025). Recent advances include reasoning-oriented models that employ RL to improve step-by-step reasoning capabilities (Sun et al., 2025; Wu et al., 2025; Li et al., 2025c). In the context of RAG, Search-R1 (Jin et al., 2025b) represents a pioneering and excellent effort to apply RL for training LLMs to dynamically interact with search engines. However, as noted in empirical studies (Jin et al., 2025a), existing RL approaches (Song et al., 2025) for reasoning-search interleaved agents face challenges in exploration efficiency and training stability.

## 3 METHOD

### 3.1 PRELIMINARY

**RAG Task Formulation** RAG addresses this limitation of LLMs when answering complex questions that require external knowledge beyond their training data. Formally, given a question $q$ and a golden answer $a$ from a dataset $\mathcal{D} = \{(q_i, a_i)\}_{i=1}^n$, the LLM alternates between generation and retrieval. At each step, it generates reasoning text or a search query, which is used to retrieve documents $d = \{d_1, d_2, \ldots, d_k\}$ from an external knowledge source $\mathcal{R}$ (*e.g.*, a search engine or database), and produces a final answer.

**RLVR Enhanced RAG** RLVR extends the RAG framework by integrating retrieval and reasoning into a reinforcement learning loop (Li et al., 2025b). The learning process is guided by a verifiable reward signal based on an objective correctness criterion, such as exact match. Formally, for each question-answer pair $(q, y)$, the reward signal $r(q, y)$ provides feedback indicating whether the generated answer satisfies predefined verification criteria.

**GRPO Algorithm** GRPO (Shao et al., 2024) is an emerging RL algorithm for training LLM policies. Formally, GRPO trains a target policy LLM $\pi_\theta$ using trajectories collected from a previous policy $\pi_{\theta_{old}}$. The goal is to maximize the expected reward while keeping the learned policy close to a fixed reference policy $\pi_{ref}$ (e.g., the pre-trained LLM prior to RL fine-tuning), ensuring training stability. For a given query $q$, GRPO generates multiple trajectories through rollouts and computes a normalized reward as the advantage. Moreover, for readability, the descriptions related to GRPO in the main text do not distinguish between $\pi_{\theta_{old}}$ and $\pi_\theta$.

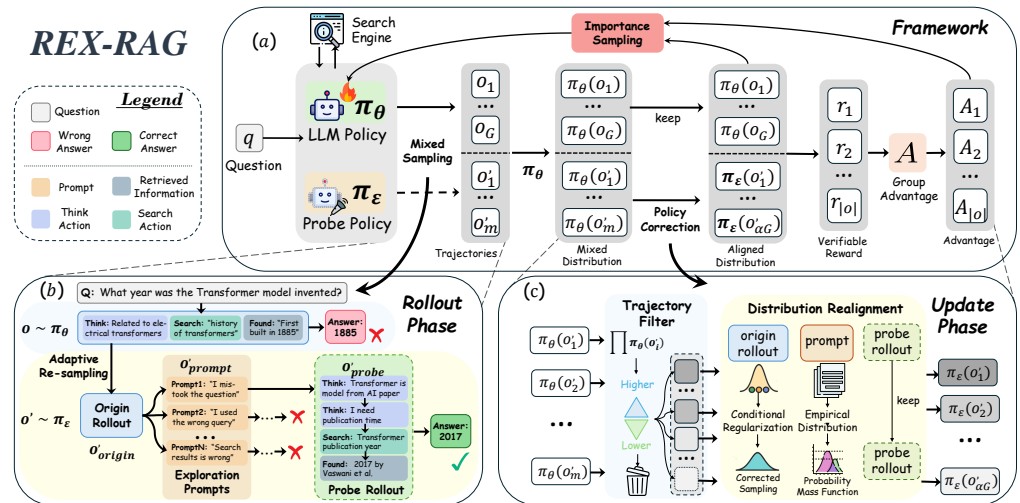

Figure 2: Overview of REX-RAG. (a) Overall framework architecture; (b) Mixed Sampling Strategy in Rollout Phase that combines policy and probe sampling; (c) Policy Correction Mechanism in Update Phase that corrects distribution shift.

## 3.2 REX-RAG FRAMEWORK

In this work, we propose REX-RAG, a novel framework that addresses the exploration challenge in RLVR-based RAG through two key innovations. As illustrated in Fig.2, during the Rollout Phase (Fig. 2 (b)), a Mixed Sampling Strategy generates diverse trajectories by combining actions from both the target policy $\pi_\theta$ and the probe policy $\pi_\varepsilon$ to escape "dead ends". In the subsequent Update Phase (Fig. 2 (c)), a Policy Correction Mechanism applies importance sampling to correct distribution shifts introduced by mixed sampling, ensuring stable policy learning while incorporating insights from exploratory rollouts.

**Framework Details** REX-RAG is implemented using GRPO (Sec. 3.1) as the underlying reinforcement learning algorithm. Regarding the prompt format, we follow the Search-R1 protocol (Jin et al., 2025b), which uses specialized tokens to define actions like searching and answering. This allows the model to autonomously interact with the search engine. The specific actions are detailed in the Appendix F. The reward function is a rule-based exact match, assigning a reward of 1 if the model's answer exactly matches the ground truth, and 0 otherwise.

## 3.3 MIXED SAMPLING STRATEGY

The Mixed Sampling Strategy enhances exploration by employing a mixed behavior policy that combines trajectories from both the current policy $\pi_\theta$ and the probe policy $\pi_\varepsilon$, thus, the mixed behavior policy can be formulated as:

$$\mu = \{\pi_\theta, \pi_\varepsilon\}. \tag{1}$$

Specifically, the strategy adaptively samples from both policies to maintain exploration diversity. It operates through a two-stages process: first sampling trajectories from the LLM policy, then adaptively performing probe sampling based on the proportion of incorrect paths.

**Adaptive Probe Re-sampling** To effectively balance exploration and exploitation, REX-RAG introduces an adaptive probe re-sampling mechanism that dynamically adjusts the degree of exploration based on the observed performance of the current policy.

The exploration process begins by sampling $n$ trajectories for each question. After collecting the corresponding rewards $\{r_1, r_2, \ldots, r_n\}$, where each $r_i \in [0, 1]$, additional exploratory trajectories are sampled in an adaptive manner. Specifically, each trajectory is resampled with probability

$p(1 - r_i)$, where $p \in [0, 1]$ is a hyperparameter that controls sampling ratio. This adaptive mechanism encourages more exploration when the policy underperforms and less when it performs well. Consequently, for each question, the expected number of resampled trajectories is given by:

$$m = p \sum_{i=1}^{n} (1 - r_i). \tag{2}$$

**Construction of Probe Policy**    To enable effective exploration, the probe policy $\pi_\varepsilon$ is constructed using a simple prompt-guided augmentation strategy, which generates exploratory trajectories by injecting exploratory guidance into the original reasoning process.

Each exploratory trajectory $o'$ is composed by concatenating three components:

$$o' = o'_{\text{origin}} \oplus o'_{\text{prompt}} \oplus o'_{\text{probe}}, \tag{3}$$

where $\oplus$ denotes sequence concatenation. As formulated in the equation, each exploratory trajectory $o'$ consists of three parts: $o'_{\text{origin}}$, which is the original model rollout up to the point where it produces an incorrect or premature answer, preserving the initial reasoning context; $o'_{\text{prompt}}$, an exploration prompt sampled from a curated prompt pool $\mathcal{P}$ designed to inject alternative reasoning directions; and $o'_{\text{probe}}$, a new continuation generated by the target model $\pi_\theta$ conditioned on the modified context.

The prompt pool $\mathcal{P}$ is constructed by rephrasing a comprehensive reflection prompt into $k$ diverse fragments using GPT-4.5 (OpenAI, 2025). Each fragment represents a distinct reasoning strategy or question reformulation, designed to stimulate exploration and diversify model behavior. The full list of base prompts and their derived fragments, as well as an empirical analysis of prompt impact, are provided in Appendix G and A.2.

### 3.4    POLICY CORRECTION MECHANISM

**Distribution Shift Chanllenge**    If the mismatch between the behavior policy $\mu = \{\pi_\theta, \pi_\varepsilon\}$ and the target policy $\pi_\theta$ introduced by the mixed sampling strategy is not addressed, model-generated samples are systematically underweighted, whereas tokens from exploration prompts are overweighted. As a result, tokens in inserted spans with negative advantages may be excessively penalized, potentially falling outside $\pi_\theta$'s support, whereas regions with positive advantages risk entropy collapse due to overly concentrated probabilities. Although GRPO's clipping trick partially addresses these issues, it does not apply during the first update in each training step, leaving the problem unresolved. Fundamentally, using an on-policy estimator in an off-policy setting introduces estimation bias and instability. For detailed mathematical analysis, refer to Appendix B.2. To mitigate this, we propose a *Policy Correction Mechanism* (Fig. 2 (c)), which reduces distribution shift and gradient bias via two steps: (i) *Trajectory Filtering*, and (ii) *Distribution Realignment*.

**Trajectory Filtering**    A trajectory filtering mechanism is first introduced to preferentially select rollouts from the probe policy that closely approximate the target policy, thereby mitigating instability and bias. Specifically, trajectories $o'$ are filtered according to their log-likelihood under the current policy $\pi_\theta$, retaining those consistent enough with it. The retention ratio is controlled by a hyperparameter $\alpha$. After filtering, for each question $t$, the retained trajectories are combined with those generated from the target policy:

$$\mathcal{O}_t = \big\{ o_i \mid o_i \sim \pi_\theta \big\}_{i=1}^{G} \ \cup \ \big\{ o'_j \mid o'_j \sim \pi_\varepsilon \big\}_{j=1}^{\alpha G}. \tag{4}$$

**Distribution Realignment**    Despite the trajectory filtering, a significant distributional mismatch still exists between the mixed behavior policy $\mu$ and the target policy $\pi_\theta$. Specifically, we first define the distribution of the Probe Policy through a principled realignment mechanism. Then, leveraging the theory of multiple importance sampling, we derive a custom optimization objective.

**Probe Policy Definition** is nontrivial because the probe policy constructs trajectories by augmenting original rollouts with injected prompts and subsequent continuations. To model $\pi_\varepsilon$ accurately, trajectories are decomposed into segments, each modeled individually under $\pi_\varepsilon$. Specifically, the prefix segment is treated as sampled from a truncated version of $\pi_\theta$ conditioned on failure, where $z$ represents the empirical failure rate. The prompt segment is deterministically selected and modeled by an

empirical probability mass function (PMF) over the prompt pool. Finally, the continuation segment is sampled directly from $\pi_\theta$ and thus requires no correction. The probe policy is thus defined as:

$$\pi_\varepsilon(o'_{i,t} \mid q_i, o'_{i<t}) = \begin{cases} \dfrac{\pi_\theta(o'_{i,t} \mid q_i, o'_{i<t})}{z^{1/|o'_{\text{origin}}|}}, & \text{if } o'_{i,t} \in o'_{\text{origin}} \\[2mm] \text{PMF}(o'_{i<t}, o'_{i,t}), & \text{if } o'_{i,t} \in o'_{\text{prompt}} \\[2mm] \pi_\theta(o'_{i,t} \mid q_i, o'_{i<t}), & \text{if } o'_{i,t} \in o'_{\text{probe}} \end{cases} \quad (5)$$

The specific design details and the construction method of the probability mass function based on frequency distribution are provided in the Appendix B.3.

**Multiple Importance Sampling** is then further employed to correct the distributional mismatch between the mixed behavior policy $\mu$, from which data is collected, and the target policy $\pi_\theta$, under which the model is optimized. The importance ratio for action $o_{i,t}$ at time step $t$ within trajectory $i$ is computed according to the balance heuristic (Veach and Guibas, 1995) as:

$$\omega_{i,t} = \frac{(1 + \alpha)\, \pi_\theta(o_{i,t} \mid q_i, o_{i,<t})}{\pi_\theta(o_{i,t} \mid q_i, o_{i,<t}) + \alpha\, \pi_\varepsilon(o_{i,t} \mid q_i, o_{i,<t})}. \quad (6)$$

The policy is then optimized with the GRPO objective:

$$J_{\text{GRPO}}(\theta) = \mathbb{E}_{\substack{q \sim \mathcal{D} \\ \{o_i\} \sim \mu(\cdot|q)}} \left[ \frac{1}{|\mathcal{O}|} \sum_{i=1}^{|\mathcal{O}|} \frac{1}{|o_i|} \sum_{t=1}^{|o_i|} \min\Big( \omega_{i,t} \hat{A}_{i,t},\, \text{clip}(\omega_{i,t}, \varepsilon) \hat{A}_{i,t} \Big) - \beta\, D_{\text{KL}}(\pi_\theta \,\|\, \pi_{\text{ref}}) \right], \quad (7)$$

where the behavior policy is updated to a mixture $\mu$, the advantage is scaled by the importance ratio from Eq. (6), and the group size is set to $|\mathcal{O}|$.

**Inference Behavior.** During the inference phase, the exploration mechanisms, including the Mixed Sampling Strategy and Policy Correction Mechanism, are deactivated. The model directly utilizes the learned policy to generate answers without any exploratory prompts or trajectory modifications, ensuring a deterministic and efficient generation process based on its training.

## 4 EXPERIMENT

We conduct extensive evaluations of REX-RAG on seven QA benchmarks, including performance improvement, ablation studies and generalizability analysis. Additional analysis in Appendix A further explores the impact of hyper-parameters and exploration prompts. The results on resampling parameter $p$ highlight sample efficiency, where a modest increase in trajectory sampling yields significant performance gains. Moreover, performance improves with a larger set of exploration prompts. Significance tests validate the statistical reliability of our findings.

### 4.1 EXPERIMENTAL SETUP

**Datasets** We evaluate REX-RAG on seven QA benchmarks: three general QA datasets NQ (Kwiatkowski et al., 2019), TrivialQA (Joshi et al., 2017), and PopQA (Mallen et al., 2023), together with four Multi-Hop QA datasets HotpotQA (Yang et al., 2018), 2WikiMultiHopQA (Ho et al., 2020), Musique (Trivedi et al., 2022), and Bamboogle (Press et al., 2023). In line with earlier studies (Jin et al., 2025b;a), we merge the NQ and HotpotQA training sets for REX-RAG training. The test splits of NQ and HotpotQA are treated as in-domain evaluations, and the remaining are used for out-of-domain evaluation. For detailed information, please refer to Appendix C.2.

**Baselines** To evaluate the effectiveness of REX-RAG, we compare it with several baselines, categorized into two groups: (1) non-fine-tuned methods, including Naive RAG (Lewis et al., 2020), IRCOT (Trivedi et al., 2023), and Search-o1 (Li et al., 2025a); and (2) fine-tuned methods, including R1-like (Guo et al., 2025) training using PPO (Schulman et al., 2017) without retrieval and those with retrieval (Jin et al., 2025b) using GRPO (Shao et al., 2024).

Table 1: Main experimental results on seven QA benchmarks. Best performance is highlighted in **bold**; the second best is underlined. ♡ denotes in-domain datasets (trained on), ◇ denotes out-of-domain datasets. All results are Exact Match Accuracy (%)

. Additional statistical analysis and significance testing are detailed in the Appendix A.3.

| Methods | General QA | | | Multi-Hop QA | | | | Avg. |
|---|---|---|---|---|---|---|---|---|
| | NQ♡ | TriviaQA◇ | PopQA◇ | HotpotQA♡ | 2wiki◇ | Musique◇ | Bamboogle◇ | |
| **Qwen2.5-3B-Base/Instruct** | | | | | | | | |
| RAG | 34.8 | 54.4 | 38.7 | 25.5 | 22.6 | 4.7 | 0.8 | 27.0 |
| IRCoT | 11.1 | 31.2 | 20.0 | 16.4 | 17.1 | 6.7 | 24.0 | 18.1 |
| Search-o1 | 23.8 | 47.2 | 26.2 | 22.1 | 21.8 | 5.4 | **32.0** | 25.5 |
| R1-base | 22.6 | 45.5 | 17.3 | 20.1 | 26.8 | 5.5 | 22.4 | 22.9 |
| R1-instruct | 21.0 | 44.9 | 17.1 | 20.8 | 27.5 | 6.0 | 19.2 | 22.4 |
| Search-R1-base | 42.1 | 58.3 | 41.3 | 29.7 | 27.4 | 6.6 | 12.8 | 31.2 |
| Search-R1-instruct | 39.7 | 56.6 | 39.1 | 33.1 | 31.0 | 12.4 | 23.2 | 33.6 |
| **REX-RAG (Ours)** | **43.9** | **60.4** | **44.2** | **37.4** | **39.7** | **14.5** | 31.2 | **38.7** |
| **Qwen2.5-7B-Base/Instruct** | | | | | | | | |
| RAG | 34.9 | 58.5 | 39.2 | 29.9 | 23.5 | 5.8 | 20.8 | 30.4 |
| IRCoT | 22.4 | 47.8 | 30.1 | 13.3 | 14.9 | 7.2 | 22.4 | 23.9 |
| Search-o1 | 15.1 | 44.3 | 13.1 | 18.7 | 17.6 | 5.8 | 29.6 | 20.6 |
| R1-base | 29.7 | 53.9 | 20.2 | 24.2 | 27.3 | 8.3 | 29.6 | 27.6 |
| R1-instruct | 27.0 | 53.7 | 19.9 | 23.7 | 29.2 | 7.2 | 29.3 | 27.1 |
| Search-R1-base | 39.5 | 56.0 | 38.8 | 32.6 | 27.0 | 12.5 | 36.0 | 35.0 |
| Search-R1-instruct | 42.9 | 62.3 | 42.7 | 38.6 | 34.6 | 16.2 | 40.0 | 39.6 |
| **REX-RAG (Ours)** | **45.5** | **62.6** | **44.3** | **42.2** | **43.7** | **19.7** | **44.8** | **43.2** |

**Implementation Details**   For external search engines, we utilize the December 2018 Wikipedia dump (Karpukhin et al., 2020) as our primary data source and employ the E5-base-v2 model (Wang et al., 2022) as the retriever. During each retrieval step, the top-3 documents returned by the retriever are provided as additional context. For REX-RAG, we adopt Qwen2.5-3B and Qwen2.5-7B as base models (Team, 2024), using GRPO as the default RL algorithm. The hyperparameters $\alpha$ and $p$ are set to default values of 0.12 and 0.2. For further details on experimental settings, please refer to the Appendix C. For evaluation, we mainly rely on the exact match. Additionally, most of the baseline results in Table 1 are taken from Search-R1 (Jin et al., 2025b;a).

## 4.2 Overall Performance

Table 1 presents main results across seven QA benchmarks. REX-RAG demonstrates consistent and substantial improvements over all baseline methods across both model sizes and dataset types.

**Performance Gains**   REX-RAG achieves significant performance improvements over the strongest baseline (Search-R1-instruct): +5.1% average improvement on Qwen2.5-3B (38.7% vs 33.6%) and +3.6% on Qwen2.5-7B (43.2% vs 39.6%). These gains are particularly pronounced on multi-hop reasoning tasks, where REX-RAG shows +8.7% improvement on 2Wiki and +4.3% on HotpotQA for the 3B model. These gains are especially high on multi-hop questions because their complex reasoning spaces demand effective exploration, and REX-RAG's probe policy excels at navigating this complexity to find optimal paths.

**Out-of-Domain Generalization**   REX-RAG also exhibits strong generalization capabilities across out-of-domain datasets. On TriviaQA, PopQA, 2Wiki, MuSiQue, and Bamboogle—none of which were seen during training—REX-RAG consistently outperforms baselines by substantial margins. This suggests that the mixed sampling strategy successfully learns generalizable reasoning patterns rather than overfitting to specific dataset characteristics.

## 4.3 Ablation Studies

### 4.3.1 Ablation on Key Components

**Component Analysis**   Table 2 presents ablation studies examining the contribution of each component in REX-RAG. We systematically remove or modify key components to understand their individual impact. (1) **Full REX-RAG**: Our complete method achieving 38.7% average performance.

Table 2: Ablation study over key components in REX-RAG (Qwen2.5-3B, GRPO).

| Methods | General QA | | | Multi-Hop QA | | | | Avg. |
|---------|------|----------|-------|----------|------|---------|-----------|------|
| | NQ | TriviaQA | PopQA | HotpotQA | 2wiki | Musique | Bamboogle | |
| **REX-RAG** | **43.9** | **60.4** | **44.2** | **37.4** | **39.7** | **14.5** | **31.2** | **38.7** |
| Coarse PPD | 45.4 | 60.9 | 44.1 | 35.4 | 35.1 | 10.7 | 23.2 | 36.4 |
| w/o IS | 45.4 | 61.8 | 43.9 | 32.5 | 28.8 | 8.1 | 13.6 | 33.4 |
| w/o TF | 39.7 | 54.2 | 36.6 | 26.0 | 26.4 | 5.5 | 9.6 | 28.2 |
| w/o IS&TF | 39.5 | 56.1 | 41.5 | 26.6 | 26.0 | 5.3 | 8.8 | 29.1 |

(2) **Coarse PPD**: Uses a simplified probe policy definition where the first token of inserted prompts are assigned probability $1/k$, while remaining prompt tokens are assigned probability 1. This leads to a 2.3% performance drop. (3) **w/o IS**: Removes importance sampling, treating all trajectories equally during training. This results in a 5.3% performance degradation. (4) **w/o TF**: Eliminates trajectory filtering, including all probe-generated trajectories regardless of quality. Performance drops by 10.5%. (5) **w/o IS&TF**: Removes the entire Policy Correction Mechanism, including IS and TS, essentially reducing to naive trajectory augmentation. This causes a 9.6% performance drop.

**Key Insights** The ablation results reveal several important insights: First, the Policy Correction Mechanism is a critical component, with its removal causing a large performance degradation. Second, trajectory filtering is essential for maintaining training stability. Without it, noisy exploratory trajectories significantly harm performance. Third, even coarse probability estimation provides substantial benefits over no correction, though precise modeling yields optimal results. These findings validate the effectiveness of our framework and design choices.

### 4.3.2 ALGORITHM GENERALIZABILITY

Table 3 demonstrates that REX-RAG's benefits generalize across different reinforcement learning algorithms. When trained with DAPO (Yu et al., 2025) instead of GRPO, REX-RAG maintains substantial improvements over Search-R1 (38.4% vs 34.8% average performance), though gains are slightly smaller than with GRPO. This suggests that REX-RAG is algorithm-agnostic and can be integrated with various RL frameworks. Interestingly, DAPO shows stronger performance on general QA tasks for Search-R1, while GRPO excels on multi-hop reasoning. REX-RAG benefits from both algorithms but shows more consistent improvements with GRPO, likely due to GRPO's group-based advantage estimation being more compatible with our mixed sampling strategy.

Table 3: Algorithm generalizability analysis comparing GRPO and DAPO frameworks on Qwen2.5-3B. Scores represent Exact Match Accuracy (%) averaged across General QA and Multi-Hop QA.

| Methods | General QA | Multi-Hop QA | Avg. |
|---------|------------|--------------|------|
| **GRPO** | | | |
| Search-R1 | 47.2 | 19.1 | 31.2 |
| REX-RAG | 49.5 | 30.7 | 38.7 |
| **DAPO** | | | |
| Search-R1 | 50.9 | 22.7 | 34.8 |
| REX-RAG | 48.4 | 30.9 | 38.4 |

### 4.4 CASE STUDIES AND VISUALIZATION

Fig. 3 presents a visualization analysis comparing reasoning trajectories of original Qwen2.5-7B against the same model enhanced with REX-RAG, using uncertainty quantification method from **LogTokU (Ma et al., 2025)**. Following the framework, we analyze **Aleatoric Uncertainty (AU)** representing inherent data randomness and **Epistemic Uncertainty (EU)** capturing model knowledge gaps through token-level confidence scoring. The visualization demonstrates that REX-RAG achieves universally higher reliability scores for reasoning tokens, with values frequently falling in the 0.6-0.8 range, whereas the baseline exhibits lower reliability (typically in the 0.2-0.4 range). This

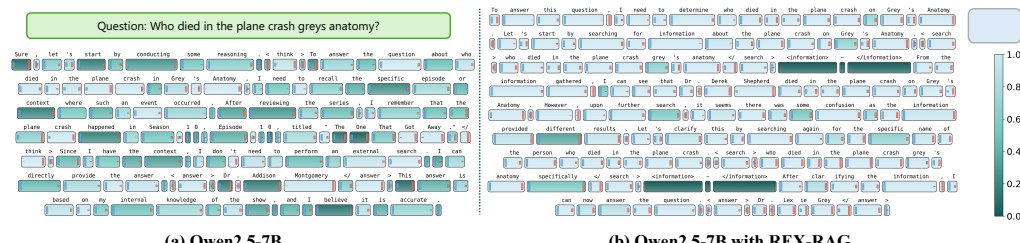

**(a) Qwen2.5-7B**        **(b) Qwen2.5-7B with REX-RAG**

Figure 3: Uncertainty quantification visualization comparing Qwen2.5-7B (left) and Qwen2.5-7B with REX-RAG (right). Color intensity represents uncertainty levels; Blue bars represent Aleatoric Uncertainty (AU) and orange bars represent Epistemic Uncertainty (EU). REX-RAG demonstrates coherent reasoning with reduced epistemic uncertainty and higher reliability scores.

indicates REX-RAG exhibits superior confidence calibration and more reliable decision-making throughout the reasoning process.

The uncertainty analysis reveals that REX-RAG exhibits high AU combined with low EU, providing evidence that REX-RAG is more exploratory precisely when it possesses relevant knowledge. This behavior demonstrates that REX-RAG's probe policy effectively identifies situations where multiple valid reasoning paths exist (high AU) while maintaining confidence in its knowledge base (low EU), leading to more thorough exploration of the solution space. In contrast, the baseline model shows the opposite pattern with low AU and high EU, indicating overconfidence in limited reasoning paths while lacking awareness of knowledge gaps.

Beyond uncertainty patterns, the visualization shows that REX-RAG produces significantly more standardized and coherent output formats compared to the baseline's fragmented and irregular response structures. This highlights that REX-RAG offers more reliable confidence estimation, coherent reasoning, and overall robustness in RAG reasoning.

While the quantitative results and uncertainty visualizations highlight REX-RAG's strengths, understanding the model's limitations is crucial. To offer deeper insights for future progress of RAG reasoning, we delve into the anatomy of failure through a detailed error case analysis in Appendix D.

## 5 LIMITATION

We discuss main limitations of our current approach; further details are provided in the Appendix E.

**Limited Exploration Strategy** Our method relies on fixed-pool prompt insertion, which, though effective, can be improved. Future work could include model-generated prompts, backtracking-based search, or full-path restructuring for more comprehensive exploration.

**Computational Overhead** The mixed sampling strategy introduces introduces a training-only overhead of $p$ additional trajectories. Though more efficient than uniform oversampling, difficulty-predictive sampling could reduce this overhead but remains challenging.

## 6 CONCLUSION

This work addresses the "dead end" problem in reinforcement learning-based retrieval-augmented generation, where models become trapped in unproductive reasoning paths during policy optimization. Our REX-RAG framework introduces the Mixed Sampling Strategy and the Policy Correction Mechanism to enable systematic exploration while maintaining training stability. Comprehensive experiments demonstrate consistent improvements over strong baselines, with particularly notable gains on multi-hop reasoning tasks. Our key contribution lies in providing a principled approach to exploration in LLM reasoning systems through importance sampling-based distributional correction. This insight may offer a practical solution for improving retrieval-augmented generation systems and provides a new exploration perspective for LLM reinforcement learning.

## 7 ETHICS STATEMENT

This research adheres to the ICLR Code of Ethics. Our work aims to enhance the reliability of LLMs by improving their factual grounding and reasoning capabilities in RAG, which can help mitigate the potential risks of misinformation and "hallucination," thereby creating a positive societal impact. We acknowledge that, like all models trained on large-scale data, the pretrained models (Qwen2.5) and data sources (e.g., Wikipedia) we use may contain existing societal biases. The outcomes of this research should therefore be used with an awareness of these inherent limitations. We intend for this work, which aims to build more accurate and dependable AI systems, to be applied in fields beneficial to society.

## 8 REPRODUCIBILITY STATEMENT

We are committed to ensuring the reproducibility of our work and will release all associated code and models publicly upon publication. As stated in the introduction, an anonymous repository has already been provided for review. To further facilitate replication, we have provided extensive experimental details throughout the paper and appendix. Specifically, we have detailed the seven public benchmarks used for evaluation and our implementation specifics (Sec. 4.1, Appendix C). Moreover, the appendices offer a complete description of the computational environment and infrastructure (Appendix C.3), a full table of hyperparameter configurations ( C.4), the instruction prompt for RL training (Appendix F.2) and the entire set of 30 exploration prompts used in our experiments (Appendix G), so that other researchers can reproduce our results.

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
