# Appendix

## CONTENTS

# A  ADDITIONAL EXPERIMENTAL ANALYSIS

## A.1  ANALYSIS OF HYPER-PARAMETERS

REX-RAG introduces a hyperparameter $p$ that controls the number of additionally sampled trajectories. As shown in the Table 4, sampling only an extra 12% of trajectories yields a substantial performance improvement over Search-R1. By contrast, Search-R1 attains only a negligible gain even when using 20% more trajectories, highlighting the superior sample efficiency of REX-RAG. Moreover, we observe a positive correlation between model performance and the resampling parameter $p$; with 20% additional sampling, the improvement becomes even more pronounced. This property allows practitioners to flexibly trade off performance gains against computational cost according to their specific needs and resource constraints.

| Sampling Strategy | General | Multi-Hop | Avg. |
|---|---|---|---|
| **Search-R1** | | | |
| 5 rollouts (+0%) | 47.2 | 19.1 | 31.2 |
| 6 rollouts (+20%) | 47.6 | 19.1 | 31.3 |
| **REX-RAG** | | | |
| 5.6 (+12% ← 12%) | 48.7 | 23.4 | 34.2 |
| 5.6 (+12% ← 20%) | 49.5 | 30.7 | 38.7 |

Table 4: Impact of trajectory sampling strategies on performance. Expected rollout counts shown for REX-RAG under maximum resampling scenarios (all initial outputs incorrect).

## A.2  ANALYSIS OF EXPLORATION PROMPT

As shown in the Table 5, we examined how varying exploration prompts affects model performance. With five prompts, we observe modest improvements on General QA and Multi-Hop QA. However, when expanding from five to thirty prompts, REX-RAG achieves a substantial performance gain relative to Serach-R1. These results indicate that the REX-RAG framework exhibits strong scalability, rather than merely benefiting from a small set of specially selected prompts.

| Sampling Strategy | General | Multi-Hop | Avg. |
|---|---|---|---|
| Search-R1 | 47.2 | 19.1 | 31.2 |
| REX-RAG(5 Prompts) | 48.3 | 20.0 | 32.1 |
| REX-RAG(30 Prompts) | 49.5 | 30.7 | 38.7 |

Table 5: Impact of Number of Exploration Prompt

## A.3  STATISTICAL ANALYSIS AND SIGNIFICANCE TEST

Given that Exact Match is a binary evaluation metric, we adopt the McNemar test to determine whether the performance differences observed in the ablation study constitute statistically significant improvements or degradations. As shown in Table 2, we evaluate a total of five models. In this subsection, we first rank the models by their Average scores in descending order and then perform pairwise comparisons between successive models.

Each numerical value in the Table 6 represents the p-value corresponding to the statistical test of the alternative hypothesis, evaluating the difference between the two models across various benchmark.

As shown in Table 6, the majority of the test results are significant (p-value $< 0.05$). While the results on a few individual benchmarks are not statistically significant, this does not affect the overall conclusions presented in the main text.

Table 6: Significance Test over key components in REX-RAG (Qwen2.5-3B,GRPO). Overall represents the results of the tests conducted on seven benchmarks. The rest are the test results obtained on each benchmark.

| Alternative Hypothesis | General QA | | | Multi-Hop QA | | | | Overall |
|---|---|---|---|---|---|---|---|---|
| | NQ | TriviaQA | PopQA | HotpotQA | 2wiki | Musique | Bamboogle | |
| REX-RAG $\neq$ Coarse PPD | $1e{-}9$ | $5e{-}15$ | $4e{-}9$ | $1e{-}50$ | $1e{-}74$ | $1e{-}10$ | $2e{-}2$ | $5e{-}144$ |
| Coarse PPD $\neq$ w/o IS | $9e{-}1$ | $1e{-}3$ | $4e{-}1$ | $2e{-}10$ | $8e{-}55$ | $1e{-}5$ | $2e{-}2$ | $3e{-}33$ |
| w/o IS $\neq$ w/o IS&IF | $3e{-}20$ | $7e{-}62$ | $5e{-}16$ | $4e{-}44$ | $8e{-}13$ | $1e{-}8$ | $1e{-}1$ | $5e{-}128$ |
| w/o IS&IF $\neq$ w/o TF | $7e{-}1$ | $3e{-}7$ | $1e{-}53$ | $1e{-}1$ | $2e{-}1$ | $7e{-}1$ | $1$ | $1e{-}24$ |

# B  MATHEMATICAL FORMULATIONS AND DERIVATIONS

## B.1  GRPO ALGORITHM

GRPO (Shao et al., 2024) is a reinforcement-learning algorithm for aligning large language models that removes the value/critic network by computing *group-relative* advantages across multiple sampled outputs for the same prompt. The baseline is the group's average reward, and policy updates are additionally regularized by a KL term to a frozen reference model.

For each prompt $q$, sample a group of $G$ outputs $\{o_i\}_{i=1}^{G}$ from the old policy $\pi_{\theta_{\text{old}}}$. Define the likelihood ratio $\rho_{i,t} = \frac{\pi_\theta(o_{i,t}|q,o_{i,<t})}{\pi_{\theta_{\text{old}}}(o_{i,t}|q,o_{i,<t})}$. GRPO maximizes: where $\varepsilon$ is the PPO clipping parameter and $\beta$ controls KL regularization to the reference policy $\pi_{\text{ref}}$.

**Outcome supervision**  Let $r_\phi$ denote a reward scoring each output. For a fixed $q$, obtain rewards $r = \{r_i\}_{i=1}^{G}$, one per output $o_i$. Compute the group mean and standard deviation

$$\mu_r = \frac{1}{G}\sum_{i=1}^{G} r_i, \qquad \sigma_r = \text{std}(r_1, \ldots, r_G).$$

Normalize each reward $\tilde{r}_i = \frac{r_i - \mu_r}{\sigma_r}$, and assign a constant advantage to all tokens in $o_i$:

$$\widehat{A}_{i,t} = \tilde{r}_i, \qquad \forall t \in \{1, \ldots, |o_i|\}. \tag{8}$$

## B.2  DISTRIBUTION SHIFT

For the sake of analytical simplicity, we disregard the clipping technique and the KL-divergence regularization term in GRPO. If we intend to employ data drawn from the mixture policy $\mu$ to optimize the target policy $\theta$, the unbiased gradient is given by:

$$\nabla_\theta J(\theta) = \mathbb{E}_{q,\{o_i\}\sim\mu}\left[\frac{1}{G}\sum_{i=1}^{G}\frac{1}{|o_i|}\sum_{t=1}^{|o_i|}\hat{A}_{i,t}\,\nabla_\theta\frac{\pi_\theta(o_{i,t}\mid q,o_{i,<t})}{\mu(o_{i,t}\mid q,o_{i,<t})}\right]. \tag{9}$$

If, instead, we apply no corrective procedure and directly use the data collected under the mixture policy $\mu$ to optimize $\theta$, the gradient we actually compute becomes:

$$\tilde{g}(\theta) = \mathbb{E}_{q,\{o_i\}\sim\mu}\left[\frac{1}{G}\sum_{i=1}^{G}\frac{1}{|o_i|}\sum_{t=1}^{|o_i|}\hat{A}_{i,t}\,\nabla_\theta\frac{\pi_\theta(o_{i,t}\mid q,o_{i,<t})}{\pi_{\theta_{\text{old}}}(o_{i,t}\mid q,o_{i,<t})}\right]. \tag{10}$$

Subtracting the two importance ratios yields the bias:

$$\Delta_{i,t} = \tilde{\rho}_{i,t} - \rho_{i,t}$$
$$= \frac{\pi_{\theta,i,t}}{\pi_{\theta_{\text{old}},i,t}} - \frac{\pi_{\theta,i,t}}{\mu_{i,t}}$$
$$= \frac{\pi_{\theta,i,t}}{\mu_{i,t}} \cdot \left( \frac{\mu_{i,t} - \pi_{\theta_{\text{old}},i,t}}{\mu_{i,t}} \right)$$
$$= \tilde{\rho}_{i,t} \left( 1 - \frac{\pi_{\theta_{\text{old}},i,t}}{\mu_{i,t}} \right). \tag{11}$$

In this expression, the first factor, $\tilde{\rho}_{i,t}$, is strictly positive and can therefore be ignored. Focusing on the sign of the second factor, we observe that for tokens generated freely by the model, $\mu$ comprises both $\pi_\theta$ and $\pi_\epsilon$, where $\pi_\epsilon$ is defined only along erroneous trajectories. Consequently, $\mu$ is smaller than $\pi_\theta$, rendering the second factor negative. Thus, for tokens sampled freely by the model, the importance ratio is biased downward, leading to systematic underestimation.

Conversely, for the segments inserted by the probe policy, the second factor is positive, conferring a systematic up-weighting. This persistent high weighting can drive the probabilities of tokens with negative advantages to decline rapidly, potentially pushing them outside the support of the policy model. Tokens with positive advantages, on the other hand, may experience rapid probability increases, thereby squeezing the probabilities of alternative tokens and inducing severe entropy collapse.

### B.3 PROBE POLICY DEFINITION

For the Probe Policy, we partition the procedure into three components according to their ordering relative to the inserted prompt: (1) the segment of the model rollout up to the point of failure; (2) the inserted prompt; and (3) the subsequent trajectory obtained by conditioning on the erroneous reasoning path and the prompt as context.

$$\pi_\varepsilon(o'_{i,t} \mid q_i, o'_{i<t}) = \begin{cases} \dfrac{\pi_\theta(o'_{i,t} \mid q_i, o'_{i<t})}{z^{1/|o'_{\text{origin}}|}}, & \text{if } o'_{i,t} \in o'_{\text{origin}} \\ \text{PMF}(o'_{i<t}, o'_{i,t}), & \text{if } o'_{i,t} \in o'_{\text{prompt}} \\ \pi_\theta(o'_{i,t} \mid q_i, o'_{i<t}), & \text{if } o'_{i,t} \in o'_{\text{probe}} \end{cases} \tag{12}$$

First, for the segment of the model rollout up to the point where an error occurs, our aim is to model the region of the original policy distribution that gives rise to failures. Within the set of all trajectories that can be sampled from the original distribution, we approximate this subset using $z$, defined as the fraction of erroneous trajectories among those sampled at the current step. This yields a distribution that is truncated relative to the original policy. To make this subset of trajectories a valid probability distribution—that is, to let "the probability mass of these trajectories fill the entire space"—we renormalize it. Accordingly, we divide the probability of each token by $z^{1/|o'_{\text{origin}}|}$ as a simple sequence-level normalization.

For the inserted prompt part, we define it based on the frequency distribution. The method induces a discrete vocabulary via a tokenizer and builds a nonparametric next-token model by aggregating, for each observed prefix $p$, the multiset of successor tokens from the corpus. Each prefix is mapped to a count vector over the vocabulary; the probability mass function is the normalized frequency. Conceptually, this is an unsmoothed, memory-based (variable-length $n$-gram) estimator that returns the empirical conditional distribution of the next token given $p$, assigning zero mass to unseen events. Specifically, the construction algorithm is as shown in Algorithm 1.

For the last part, since we do not impose any restrictions on the sampling of these parts, we directly use the probability of the original policy model as the probability of the probe policy.

---

**Algorithm 1:** PMF Construction via Frequency Distribution

---

**Input:** Tokenizer $\mathcal{T}$; Prompt set $\mathcal{P} = \{s_1, \ldots, s_m\}$
**Output:** Function $\text{PMF}(p, x)$
$K \leftarrow \{\mathcal{T}(s) \mid s \in \mathcal{P}\}$;
// tokenize every prompt
$V \leftarrow$ unique tokens in $K$;
// initialize vocabulary
**foreach** $k \in K$ **do**
    **for** $i \leftarrow 0$ **to** $|k| - 1$ **do**
        $p \leftarrow k_{0:i}$;
        $C[p] \leftarrow \mathbf{0}_{|V|}$;
        // initialize frequency distribution

**foreach** $k \in K$ **do**
    **for** $i \leftarrow 0$ **to** $|k| - 1$ **do**
        $p \leftarrow k_{0:i}$; $x \leftarrow k_{i+1}$;
        $C[p][\text{V.index}(x)] \leftarrow C[p][\text{V.index}(x)] + 1$;

**Function** *PMF(p, x)*:
    counts $\leftarrow C[p]$;
    **return** $\frac{counts[\text{V.index}(x)]}{\sum counts}$;

**return** PMF;
// expose the query function to the caller

---

### B.4 COEFFICIENT FOR IMPORTANCE SAMPLING

Let the goal be to estimate the policy gradient using a mixed policy $\mu = \{\pi_\theta, \pi_\epsilon\}$. During sampling, a fraction of $\frac{1}{1+\alpha}$ of the trajectories come from $\pi_\theta$, while a fraction of $\frac{\alpha}{1+\alpha}$ of the trajectories come from $\pi_\epsilon$:

$$c_\theta = \frac{1}{1 + \alpha}, \qquad c_\epsilon = \frac{\alpha}{1 + \alpha}. \tag{13}$$

Under the *balance heuristic* (Veach and Guibas, 1995), the weight is

$$\hat{\omega}_i(x) = \frac{c_i\, p_i(x)}{\sum_j c_j p_j(x)}. \tag{14}$$

Substitute the variables into it respectively, and we can obtain the Importance ratio for estimating the policy gradient of Multiple Importance Sampling:

$$\omega = \frac{(1 + \alpha)\, \pi_\theta}{\pi_\theta + \alpha\, \pi_\varepsilon}. \tag{15}$$

## C EXPERIMENTAL IMPLEMENTATION DETAILS

### C.1 BASELINE METHODS

We evaluate REX-RAG against two categories of baselines: (1) non-fine-tuned methods, including Naive RAG (Lewis et al., 2020), IRCOT (Trivedi et al., 2023), and Search-o1 (Li et al., 2025a); and (2) fine-tuned methods, including R1-like (Guo et al., 2025) trained with PPO (Jin et al., 2025b) (with and without retrieval) using GRPO (Shao et al., 2024).

**Naive RAG** (Lewis et al., 2020) is the standard retrieval-augmented generation approach that retrieves documents using dense passage retrieval and generates answers conditioned on both the query and the retrieved context. It employs a bi-encoder architecture and marginalizes over retrieved documents during generation, enabling dynamic access to external knowledge and reducing hallucination in knowledge-intensive tasks.

**IRCOT** (Trivedi et al., 2023) interleaves reasoning and retrieval steps, alternating between generating intermediate reasoning steps and retrieving new information. This few-shot prompting approach enables step-wise information gathering and supports multi-hop reasoning by refining retrieval based on the evolving reasoning chain.

**Search-o1** (Li et al., 2025a) enhances LLM reasoning by integrating web search. It uses multi-step reasoning to analyze queries, formulate searches, and synthesize results. Iterative search-query reformulation and result ranking improve retrieval quality. The approach relies on chain-of-thought reasoning to generate comprehensive answers using diverse sources.

**R1-like Training** (Guo et al., 2025) employs RLHF via PPO to fine-tune LLMs for reasoning tasks without retrieval. Following DeepSeek-R1, it includes supervised reasoning trace training, reward modeling, and PPO optimization. This pipeline enhances reasoning quality using curated datasets and human feedback, serving as a strong non-retrieval baseline.

**Search-R1** (Jin et al., 2025b) extends R1-style training by integrating retrieval actions into the policy optimization process using GRPO. It jointly optimizes reasoning and retrieval quality, with rewards based on final answer accuracy and coherence. Retrieval is treated as part of the trajectory, allowing the model to learn effective information-seeking strategies. This serves as a strong prior baseline for evaluating the improvements brought by our proposed policy realignment mechanisms.

## C.2 DATASET DESCRIPTIONS

We evaluate REX-RAG on seven QA benchmarks: three general QA datasets NQ (Kwiatkowski et al., 2019), TrivialQA (Joshi et al., 2017), and PopQA (Mallen et al., 2023), together with four Multi-Hop QA datasets HotpotQA (Yang et al., 2018), 2WikiMultiHopQA (Ho et al., 2020), Musique (Trivedi et al., 2022), and Bamboogle (Press et al., 2023). In line with earlier studies (Jin et al., 2025b;a), we merge the NQ and HotpotQA training sets for REX-RAG training. The test splits of NQ and HotpotQA are treated as in-domain evaluations, and the remaining five datasets are used for out-of-domain evaluation.

**Natural Questions (NQ)** (Kwiatkowski et al., 2019) is a large-scale dataset featuring real Google Search queries paired with Wikipedia passages containing the answers. It includes over 300K naturally occurring questions, each annotated with both a long answer (usually a paragraph) and a short answer (typically a phrase). NQ reflects realistic information-seeking behavior across diverse topics such as history, science, and current events, with varying complexity. We use it as an in-domain benchmark, as it contributes to REX-RAG's training.

**TriviaQA** (Joshi et al., 2017) is a reading comprehension dataset containing over 95K question-answer pairs sourced from trivia websites and paired with evidence documents from Wikipedia and the web. Not all documents are guaranteed to contain the answer, requiring models to perform effective retrieval. The questions emphasize factual knowledge, making the dataset ideal for evaluating retrieval-augmented systems.

**PopQA** (Mallen et al., 2023) targets popular factual questions about widely known topics such as celebrities, movies, and sports events. It evaluates models' ability to answer questions about current and trending topics that may not appear in training corpora, highlighting the importance of real-time retrieval for up-to-date knowledge.

**HotpotQA** (Yang et al., 2018) is a multi-hop QA dataset with over 113K Wikipedia-based examples, where each question requires reasoning across at least two paragraphs. It includes bridge and comparison questions and provides supporting facts. As an in-domain benchmark, it plays a key role in evaluating REX-RAG's multi-hop reasoning performance.

**2WikiMultiHopQA** (Ho et al., 2020) extends multi-hop QA by requiring reasoning over two Wikipedia articles using varied operations like numerical, logical, and compositional reasoning. Each question involves exactly two hops and is annotated with reasoning paths and supporting evidence, facilitating fine-grained evaluation of multi-step reasoning.

**MuSiQue** (Trivedi et al., 2022) focuses on compositional multi-hop reasoning across multiple documents. Questions often involve temporal or relational reasoning and require synthesizing scattered information. It includes both answerable and unanswerable questions, testing models' ability to detect insufficient context.

**Bamboogle** (Press et al., 2023) is a challenging multi-hop QA benchmark designed to stress-test reasoning capabilities. Questions involve complex inference steps, including temporal and causal reasoning, often under ambiguous or incomplete information. It highlights the limitations of current QA systems and the need for more advanced reasoning strategies.

### C.3 COMPUTATIONAL ENVIRONMENT AND INFRASTRUCTURE

All experiments in this study were conducted on a cluster of 8 NVIDIA A800 80GB GPUs, providing the computational resources necessary for large-scale reinforcement learning training and evaluation of retrieval-augmented generation systems.

**Reinforcement Learning Framework.** We implemented our REX-RAG training pipeline using VERL (Sheng et al., 2024), an open-source distributed reinforcement learning framework developed by ByteDance for efficient large language model training. VERL is specifically designed to handle the computational challenges of RLHF at scale, providing optimized implementations of policy optimization algorithms such as PPO and GRPO.

**Retrieval Infrastructure.** Our retrieval system is built upon FAISS (Facebook AI Similarity Search) (Johnson, Douze, and Jégou, 2019) for efficient similarity search and indexing. We employ the E5 embedding model (Wang et al., 2022) to encode both queries and documents into dense vector representations, enabling semantic similarity matching for retrieval operations. The knowledge base consists of Wikipedia passages from the DPR corpus (Karpukhin et al., 2020), specifically the Wiki-18 dataset. The entire retrieval system is deployed using FastAPI.

**Data Processing and Evaluation Pipeline.** For data preprocessing, evaluation metrics computation, and baseline comparisons, we adopted the experimental framework from Search-R1 (Jin et al., 2025b). This includes standardized data loading procedures, question-answer pair formatting, retrieval corpus preparation, and evaluation protocols that ensure fair comparison across different methods. The Search-R1 framework provides implementations for computing exact match accuracy for multi-hop reasoning evaluation.

**Prompt Generation and Template Management.** We utilized GPT-4.5 for generating high-quality prompts and reasoning templates used throughout our experiments. This mainly includes the generation of exploration prompts for policy training, as shown in Appendix G.

### C.4 HYPER-PARAMETER CONFIGURATION AND TUNING

Table 7: Primary hyperparameters used by REX-RAG. Performance-related parameters were tuned for optimal GPU utilization, while other parameters follow Search R1 baseline configuration.

| Category | Hyperparameter | Value |
|---|---|---|
| **Performance** | Training Batch Size | 512 |
| | Mini Batch Size | 256 |
| | Max Token Length | 24,000 |
| | GPU Memory Utilization | 0.8 |
| | Max Batched Tokens | 8,192 |
| | Max Sequences per Batch | 1,024 |
| **Training** | Actor Learning Rate | $1 \times 10^{-6}$ |
| | Warmup Steps Ratio | 0.285 |
| | Weight Decay | 0.01 |
| | PPO Epochs | 1 |
| **Policy** | Clip Ratio | 0.2 |
| | KL Coefficient | 0.001 |
| | Use Dynamic Batch Size | True |
| **Generation** | Max Search Turns | 5 |
| | Response Length | 500 |
| | Temperature | 1.0 |
| | Top-p Value | 1.0 |

Our hyperparameter configuration strategy primarily focused on tuning algorithm-agnostic parameters that optimize GPU computational performance, particularly those related to macro batch size and GPU utilization settings. This approach ensures efficient resource utilization while maintaining training stability. For all other hyperparameters not directly related to computational performance, we maintained consistency with the Search R1 baseline configuration to ensure fair comparison and reproducibility. Table 7 presents the key hyperparameters used in our REX-RAG implementation.

The performance-oriented hyperparameters in the first category were specifically tuned to optimize computational efficiency on our hardware configuration. The training batch size of 512 and PPO mini batch size of 256 were selected to maximize throughput while maintaining gradient stability. The GPU memory utilization of 0.8 ensures efficient memory usage without risking out-of-memory errors during training. The token length and sequence limits were configured to balance between accommodating longer reasoning chains and maintaining computational feasibility.

To support deeper reasoning and allow recovery from failed attempts, we increased the Max Search Turns from 2 (as used in Search-R1) to 5. This extension enables the model to conduct further retrieval after initial errors, which is essential for our exploration-driven method. Notably, we kept the overall maximum token budget unchanged, ensuring that this change does not introduce significant additional computational overhead.

All remaining hyperparameters, including learning rates, regularization coefficients, and generation parameters, were kept consistent with the Search-R1 baseline to ensure that performance improvements can be attributed to our proposed REX-RAG methodology rather than hyperparameter optimization advantages.

## D    ERROR CASE ANALYSIS

In this section, we conduct a detailed analysis of several challenging cases where our model failed. By dissecting these failures, we aim to provide valuable insights into the persistent challenges of complex question answering and highlight promising directions for future research. We categorize these errors into three main types: Failures in Entity Disambiguation and Errors in Logical Reasoning. Our analysis reveals that while REX-RAG significantly improves performance, these edge cases represent the next frontier of challenges.

### D.1    FAILURES IN ENTITY DISAMBIGUATION

---

**Case 1: Incorrect Entity Association**

**Input:** Who is the uncle of Nicaea Of Macedon?
**Model Output:** Phalion
**Ground Truth:** Cassander

**Model's Reasoning Process:**

```
<think> To answer this question, I need to determine who the uncle of Nicaea of Macedon
    ↪ is.
<search> Nicaea of Macedon </search>
<information> Nicaea of Macedon was a daughter of the powerful regent Antipater. </
    ↪ information>
<think> To find her uncle, I need to search for Antipater's brothers. </think>
<search> Antipater brothers </search>
<information> Antipater the Idumaean ... had a brother named Phalion... </information>
<think> Based on the information, Antipater had a brother named Phalion. Therefore, the
    ↪  uncle is Phalion. </think>
<answer> Phalion </answer>
```

**Analysis:** The model correctly identified that Nicaea was the daughter of Antipater. However, the query "Antipater brothers" returned information for *Antipater the Idumaean* (father of Herod the Great), a more prominent historical figure in the search index, instead of *Antipater the Macedonian regent*. The model incorrectly associated Nicaea with the wrong Antipater and thus identified the wrong brother (Phalion). The correct entity, Antipater the regent, had a son named Cassander, who would be Nicaea's brother, not uncle. This highlights a key challenge: even if the model knows what to look for (Antipater's brother), the search results can be misleading.

---

Among the observed failure modes, Failures in Entity Disambiguation are the most common. This error occurs when a model is unable to distinguish between entities that share the same or similar names, leading to the retrieval of incorrect information. This issue is particularly damaging in

multi-hop reasoning scenarios, where an early disambiguation error can derail the entire subsequent reasoning chain. The iterative retrieval mechanism in REX-RAG is designed to mitigate this by allowing the model to refine its understanding. However, when initial search results are overwhelmingly dominated by a more famous but incorrect entity, the model can still be led astray.

## D.2 ERRORS IN LOGICAL REASONING

A second category of failure involves errors in the logical reasoning step, which occurs after all necessary information has been correctly retrieved. In these instances, the model successfully gathers the right facts but falters when performing a final comparison or logical deduction (e.g., comparing dates or numbers). While REX-RAG pushes performance by improving information retrieval and enhancing logical consistency, the bottleneck of model's intrinsic reasoning capability causes occasional failures even when information retrieval is perfect.

---

**Case 2: Error in Final Comparison**

**Input:** Which film whose director is younger, Dagudu Moothalu or Port Of Seven Seas?
**Model Output:** Port Of Seven Seas
**Ground Truth:** Dagudu Moothalu

**Model's Reasoning Process:**

```
<think> I need to find the directors of both films and their ages. </think>
<search> Dagudu Moothalu director </search>
<information> Dagudu Moothalu (1964) ... directed by Adurthi Subba Rao. </information>
<search> Port Of Seven Seas director </search>
<information> Port of Seven Seas (1938) ... directed by James Whale. </information>
<search> Adurthi Subba Rao age </search>
<information> Adurthi Subba Rao (16 December 1912 – 1 October 1975) </information>
<search> James Whale age </search>
<information> James Whale ... suicide ... on 29 May 1957 at the age of 67. [born in
    ↪ 1889] </information>
<think> Adurthi Subba Rao was born in 1912, and James Whale was born in 1889. Therefore
    ↪ , James Whale is younger. </think>
<answer> Port Of Seven Seas </answer>
```

**Analysis:** The model correctly executed a complex query plan: it identified both directors (Adurthi Subba Rao and James Whale) and retrieved their birth years (1912 and 1889, respectively). However, it failed at the final step of comparing the two dates, incorrectly concluding that a person born in 1889 is younger than one born in 1912. This is a pure reasoning failure. Another case exhibited a similar flaw, where the model correctly found George E. Mahoney's birth year (1901) and Paulo Roberto Falcão's (1953) but wrongly concluded that Mahoney was younger. These cases show that even when the RAG component works perfectly, the core reasoning capacity of the LLM can be fragile.

---

# E   LIMITATIONS, DISCUSSION, AND FUTURE WORK

**Limited Exploration Strategy**   Our current exploration mechanism relies on a relatively simple strategy—injecting prompts from a pre-constructed prompt pool to guide the model toward alternative reasoning paths. While effective, this approach may fall short of the full potential of more sophisticated exploration techniques. From the prompt perspective, online generation of exploration prompts conditioned on the model's current reasoning state may offer greater adaptivity and contextual relevance than our static prompt set. From the policy perspective, incorporating more structured search procedures, such as backtracking trees or trajectory-level search algorithms, could enable more systematic exploration across the reasoning space. Moreover, our method emphasizes local trajectory perturbation via prompt insertion, rather than global restructuring of the reasoning path. Despite these limitations, our results demonstrate that end-to-end optimization under an exploratory policy is both feasible and beneficial, laying the groundwork for future work on more principled and expressive exploration strategies.

**Computational Overhead and Adaptive Sampling Limitations**   The mixed sampling strategy inherently introduces computational overhead compared to standard policy optimization approaches. Our resampling mechanism requires a two-stage process: first performing normal sampling to assess question difficulty through initial trajectory evaluation, then conducting exploratory sampling based on the observed failure rates. This sequential approach increases computational complexity as it necessitates generating $(1 - \alpha)G$ additional exploratory trajectories from the probe policy $\pi_\varepsilon$, resulting in approximately 12% more trajectory sampling in our experiments. While this overhead

is substantially more efficient than uniform oversampling approaches (which require 20% additional trajectories for minimal gains), the computational cost scales linearly with the resampling parameter $p$ and the exploration ratio $\alpha$. A more efficient approach would involve predicting question difficulty a priori and automatically adjusting sampling quantities accordingly, eliminating the need for the initial sampling phase. However, developing reliable difficulty prediction mechanisms remains an open challenge. Furthermore, the policy realignment mechanism requires computing importance sampling ratios for each token, adding non-negligible computational complexity during training.

**Lack of Validation in Broader Agentic Tasks**  While REX-RAG demonstrates consistent improvements across seven open-domain question answering datasets, its effectiveness has only been validated within the RAG (retrieval-augmented generation) framework. Our method specifically targets reasoning-intensive QA tasks where external information retrieval and multi-turn reasoning are tightly coupled. As such, it remains unclear whether the proposed exploration and policy realignment strategies generalize to broader agentic scenarios—such as tool use, web navigation, or embodied planning—where action spaces, environmental feedback, and task dynamics differ substantially. Extending our framework to these settings would require adapting both the structured interaction protocol and the rollout mechanism to accommodate more complex state-action transitions. Future work may explore the applicability of REX-RAG's core ideas beyond QA, investigating how exploration with distribution correction can benefit general-purpose decision-making agents.

**Simplistic Trigger Mechanism**  To clearly isolate and evaluate the core contribution, REX-RAG intentionall adopt a straightforward "Exact Match" criterion to trigger exploration. While this binary decision framework provides a clear and interpretable baseline for our experiments, it does not capture the full spectrum of reasoning quality. For instance, it may overlooks flawed reasoning paths that happen to yield a correct answer, thereby missing valuable learning opportunities, and it incorrectly penalizes responses that are semantically equivalent to the ground truth but differ in phrasing. Future work should therefore focus on developing more intelligent triggers. This could involve integrating semantic similarity scores, model confidence levels, and specialized error classifiers. Furthermore, leveraging uncertainty quantification based on the model's internal state would enable a more discerning and efficient exploration strategy, maximizing learning while minimizing computational cost.

## F  STRUCTURED SEARCH INTERACTION PROTOCOL

The structured search interaction protocol employed in REX-RAG follows the framework established by Search-R1 (Jin et al., 2025b), which defines a systematic approach for integrating reasoning and retrieval operations through specialized tokens and prompt templates. The structured interaction protocol relies on four primary special tokens that delineate different phases of the reasoning and retrieval process:

### F.1  SPECIAL TOKENS

`<think>` and `</think>` encapsulate the model's internal reasoning process, allowing it to engage in chain-of-thought reasoning without external interference. Within these tags, the model can perform logical deduction, analyze given information, identify knowledge gaps, and plan subsequent actions. This internal reasoning phase is crucial for determining when external retrieval is necessary and formulating appropriate search queries.

`<search>` and `</search>` trigger external information retrieval operations. When the model generates these tokens, the content within them is interpreted as a search query that is executed against the external knowledge base. This mechanism allows for dynamic knowledge acquisition during the reasoning process.

`<information>` and `</information>` contain the retrieved external knowledge that is returned by the search engine in response to search queries. This mechanism allows for dynamic knowledge acquisition during the reasoning process.

`<information>` and `</information>` contain the retrieved external knowledge that is returned by the search engine in response to search queries. The content within these tags represents

the top search results that are automatically inserted into the model's context after a search operation. This information serves as additional context that the model can analyze and incorporate into itse content within these tags represents the top search results that are automatically inserted into the model's context after a search operation. This information serves as additional context that the model can analyze and incorporate into its reasoning process.

These special tokens serve multiple purposes: they provide clear demarcation between different operational phases, enable selective training on specific components of the reasoning process, and facilitate systematic evaluation of reasoning quality versus retrieval effectiveness.

**<answer> and </answer>** mark the final response generation phase, where the model synthesizes information from both its internal reasoning and retrieved external knowledge to produce a comprehensive answer. The content within these tags represents the model's final output, incorporating insights gained througing and retrieval process.

These special tokens serve multiple purposes: they provide clear demarcation between different operational phases, enable selective training on specific components of the reasoning process, and facilitate systematic evaluation of reasoning quality versus retrieval effectiveness.

## F.2 INSTRUCTION PROMPTS

The prompt template structure orchestrates the interaction between reasoning and retrieval components through a carefully designed format that guides the model's behavior throughout the question-answering process. The template follows a think-search-answer paradigm that promotes systematic problem-solving. The entire prompt template is demonstrated in Fig. 4.

Figure 4: Complete prompt template and example interaction for the structured search protocol

**Structured Search Interaction Protocol**

**Prompt Template:**

*Answer the given question. You must conduct reasoning inside <think> and </think> first every time you get new information. After reasoning, if you find you lack some knowledge, you can call a search engine by <search> query </search>, and it will return the top searched results between <information> and </information>. You can search as many times as you want. If you find no further external knowledge needed, you can directly provide the answer inside <answer> and </answer> without detailed illustrations. For example, <answer> xxx </answer>. Question: [question]*

**Example Interaction:**

**Question:** What is the capital of France and when was it founded?
<think> I need to find information about Paris, the capital of France, and its founding date. </think>
<search> Paris France capital founding date history </search>
<information> Paris is the capital of France. The city was founded in the 3rd century BC by the Parisii tribe... </information>
<think> Based on the retrieved information, I now have the answer to both parts of the question. </think>
<answer> The capital of France is Paris, which was founded in the 3rd century BC. </answer>

## G REVISION PROMPTS AND EXAMPLES

The revision prompts are formulated to express uncertainty and encourage critical self-evaluation without being overly prescriptive. Prompts are designed to maintain the natural reasoning flow while introducing a reflective pause that can lead to error correction and improved reasoning quality.

The Table 8 presents all 30 revision prompts used in our implementation. These prompts are randomly selected during training to provide diverse expressions of self-doubt and reflection.

Table 8: Complete collection of revision prompts used in REX-RAG for triggering self-reflection during reasoning

| ID | Revision Prompt Text | ID | Revision Prompt Text |
|---|---|---|---|
| 0 | <think> Perhaps I've overlooked critical points or slipped up in my logic. | 15 | <think> Concerned I might have overlooked key aspects or made subtle errors. |
| 1 | <think> I wonder if vital information escaped my notice or if I made an error. | 16 | <think> I might have unintentionally ignored essential details or misunderstood something. |
| 2 | <think> There might be key gaps in my understanding or errors in reasoning. | 17 | <think> Revisiting carefully, perhaps errors or oversights went unnoticed earlier. |
| 3 | <think> It's possible I've missed something important or misunderstood crucial details. | 18 | <think> Maybe important points slipped my attention, or I made a miscalculation. |
| 4 | <think> I suspect errors crept in, or essential points went unnoticed. | 19 | <think> It's likely I've overlooked something crucial or stumbled in logic. |
| 5 | <think> Maybe I've misjudged something important or neglected key facts. | 20 | <think> Reflecting, I could've missed critical clues or made errors in judgment. |
| 6 | <think> Reflecting now, I might have overlooked critical data or erred somewhere. | 21 | <think> Possibly, I misunderstood something fundamental or missed key evidence. |
| 7 | <think> Possibly, I've missed significant insights or made a mistake. | 22 | <think> Concerned about potential unnoticed mistakes or overlooked essential details. |
| 8 | <think> I'm sensing a gap or error might be present in my recent reasoning. | 23 | <think> Perhaps my earlier step wasn't entirely accurate or lacked vital points. |
| 9 | <think> I could have misinterpreted important facts or overlooked necessary details. | 24 | <think> It's conceivable that I've neglected critical information or erred. |
| 10 | <think> Aware that my reasoning might be flawed or lacking crucial points. | 25 | <think> Wondering if I've mistakenly dismissed something important or misunderstood it. |
| 11 | <think> I need to reconsider—I might've skipped vital information or erred. | 26 | <think> Maybe my previous reasoning has blind spots or unnoticed errors. |
| 12 | <think> There's a chance my previous thinking has unnoticed mistakes or omissions. | 27 | <think> I'm doubting if crucial points were missed or mistakes made earlier. |
| 13 | <think> I feel there might be something critical I overlooked or misunderstood. | 28 | <think> Feeling uncertain—perhaps critical details slipped past or were misunderstood. |
| 14 | <think> Perhaps my earlier reasoning has hidden mistakes or missing information. | 29 | <think> Recognizing possible gaps or missteps I didn't previously notice. |

# H  USAGE OF LLM

**Writing Assistance**  LLMs are employed to assist in the writing and refinement of this manuscript. This included tasks such as proofreading for grammatical errors, improving sentence structure for clarity, and rephrasing content to enhance readability. It is important to note that all AI-generated text is thoroughly reviewed, critically evaluated, and edited by the authors to ensure the accuracy and integrity of the final content. The authors take full responsibility for all statements and claims made in this paper.

**Code Implementation**  LLMs are used as a tool to facilitate the implementation of algorithms and data processing scripts. This involves generating boilerplate code, suggesting solutions for specific programming challenges, and debugging. All code generated by LLMs is manually verified and tested by the authors to ensure its correctness, efficiency, and adherence to the project's requirements.

**Research Applications**  Beyond supporting tasks, LLMs are integral to the research itself, serving multiple functions as detailed throughout the paper. These applications include acting as the base model for our experiments, refining and rephrasing prompts to guide model behavior, and other research-specific uses that are explicitly mentioned in the relevant sections of this work.