# OpenReview forum: "REX-RAG: Reasoning Exploration with Policy Correction in Retrieval-Augmented Generation"
_ICLR.cc/2026/Conference — ICLR 2026 Conference Withdrawn Submission_

### Official Review · Reviewer_4hhY · 2025-10-28

**Soundness:** 3
**Presentation:** 3
**Contribution:** 2
**Rating:** 6
**Confidence:** 3

**Summary:**

During reinforcement learning (RL) training, “dead ends” — consistently failing trajectories — often occur. Simple self-reflection does not effectively address this issue, as it tends to only slightly perturb the original path without resolving the underlying problem. To mitigate this, the proposed method introduces several design considerations. First, it employs a mixed sampling strategy, which samples from both the current policy and a probe policy. Second, it includes a policy correction mechanism to account for the distribution shift introduced by exploration.

**Strengths:**

1. The paper is well-motivated, clearly identifying a key limitation of existing self-reflection methods and proposing a principled approach to address it.

2. The paper is clearly written and easy to follow, with well-structured explanations of both intuition and methodology.

**Weaknesses:**

1. It is unclear whether the prompts used in “Construction of the Probe Policy” were also used as the prompts for the self-reflection baseline shown in Figure 1. Additionally, does the self-reflection baseline involve training with self-reflected responses, or is it only evaluated with self-reflected responses without training? This distinction is important because the main difference between the proposed mixed sampling strategy and prior self-reflection approaches is not entirely clear. Some previous works also correct answers using self-reflection and incorporate these corrected responses during training. Alternatively, is this work applying self-reflection in parallel to the sampled generations within GRPO? This needs to be clarified in the paper.

2. The experimental section lacks a comparison with existing self-reflection baselines, which are necessary to contextualize the improvements.

3. The ablation study only investigates components within the policy correction mechanism. It should also include an ablation study evaluating the contribution of the mixed sampling strategy.

**Questions:**

1. Why does this work focus solely on the RAG problem? The overall pipeline appears to be general and potentially applicable to a broader range of tasks.

2. Please provide the clear distinction with existing self-reflection works (especially those using self-reflection in training).

3. Please provide the ablation study results of the full pipeline.

---

> ### Author Response · Authors · 2025-11-21
> **Response to Reviewer 4hhY**
>
> We sincerely thank you for the constructive feedback. We appreciate your recognition that our method is "well-motivated" and "principled."
>
> We realized from your comments that we did not explicitly distinguish between test-time and training-time reflection in the initial draft, which led to confusion regarding the baselines. We provide detailed clarifications below.
>
> ---
>
> # Response to W1
>
> We apologize for the ambiguity regarding the baseline setup. To address your concerns precisely, we have broken down your comment into four specific questions.
>
> ### **(q1) Were the prompts in "Probe Policy" used for the self-reflection baseline in Figure 1?**
>
> **No**, they are different.
> - **Self-Reflection Baseline (Fig. 1)**: Following the Search-R1 protocol, this baseline uses a **single, static trigger** (e.g., "My action is not correct. Let me rethink.") to induce reflection.
> - **REX-RAG (Our Method)**: We utilize a Prompt Pool containing **30+ diverse exploration prompts** (e.g., "I might have unintentionally ignored essential details", see Appendix.G). A prompt is randomly sampled from this pool for each exploration attempt. This diversity is a key design choice to prevent the mode collapse often observed with static prompts.
>
> ### **(q2) Does the baseline involve training or just evaluation?**
>
> The "Self-Reflection" baseline in Figure 1 and throughout our paper **refers to Training-time Self-Reflection**. We align with the Search-R1 and DeepSeek-R1 methodologies, where the model is explicitly trained on trajectories that contain these self-reflected responses.
>
> ### **(q3) What is the distinction between Mixed Sampling and prior self-reflection works?**
>
> Our method differs from these standard self-reflection approaches in two critical dimensions:
> 1. **Exploration Diversity via Prompt Pool**: Unlike prior works that rely on a single, static trigger, REX-RAG utilizes a Prompt Pool (Appendix.G) containing 30+ diverse reflective prompts. This forces the model to explore the solution space from various semantic angles.
> 2. **Mathematical Rigor via Policy Correction**: This is our key contribution. Standard self-reflection simply injects prompts and trains on the generated data, ignoring the fact that these prompts make the data off-policy. REX-RAG introduces the Policy Correction Mechanism (Eq. 6) to mathematically correct the distribution shift caused by this exploration. This ensures unbiased policy optimization, which naive self-reflection lacks.
>
> ### **(q4) Is self-reflection applied in parallel to sampled generations within GRPO?**
>
> **Yes.** We implement a Mixed Sampling Strategy within the GRPO framework. For a given question, we generate a batch of trajectories that includes **both standard rollouts and exploration rollouts**. These are pooled together for the update, with the exploration trajectories re-weighted by our correction mechanism.
>
> ---
>
> # Response to W2
>
> We fully agree with the reviewer that contextualizing our improvements against existing self-reflection methods is essential.
> In our experiments, we selected Search-R1 specifically to serve as this representative baseline, as it **implements the state-of-the-art training-time self-reflection** (appending a static "rethink" trigger upon failure). We apologize if we did not sufficiently emphasize this connection in the main text.
>
> The results in Table 1 show that REX-RAG **achieves a 5.1% improvement** over this strong self-reflection baseline. We believe this comparison effectively isolates the value of our approach: moving beyond static reflection triggers to a more principled framework of diverse exploration and mathematical correction. We will update the Related Work and Experiments sections to explicitly position Search-R1 as the "Standard Self-Reflection" baseline to make this comparison clearer.
>
> ---
>
> # Response to W3
>
> We appreciate this suggestion. However, we would like to clarify the **architectural dependency inherent** in our framework.
>
> The Policy Correction Mechanism (Importance Sampling & Trajectory Filtering) is designed specifically to rectify the distribution shift introduced by the Mixed Sampling Strategy (the probe policy). Consequently, it is **structurally impossible to remove Mixed Sampling while retaining Policy Correction**.
>
> Therefore, **removing the Mixed Sampling Strategy deactivates the entire REX-RAG machinery**, causing the model to revert mathematically to the Search-R1 baseline. Thus, the comparison in Table 1 (REX-RAG vs. Search-R1) serves as the functional ablation study for the Mixed Sampling Strategy itself.
>
> We acknowledge that this logical connection was implicit in our initial draft. We will **revise Section 4.3** to explicitly frame the Search-R1 performance as the "w/o Mixed Sampling & Policy Correction" lower bound, ensuring the completeness of the component analysis is clear to the reader.

---

> > ### Author Response · Authors · 2025-11-21
> >
> > # Response to Q1
> >
> > We appreciate this insightful observation. We selected Retrieval-Augmented Generation (RAG) as our initial testbed because it presents a distinct and challenging constraint: **if the model fails to retrieve relevant documents, the reasoning process hits a definitive "dead end"** that is impossible to recover from without active exploration.
> >
> > As exemplified by the **Search-R1 failure case** detailed in our **Response to Reviewer DgE2 Q7**, agents frequently become trapped in repetitive loops of identical search queries without gaining new information. This makes RAG an ideal environment to rigorously validate the effectiveness of our exploration mechanism.
> >
> > Furthermore, we fully agree that the REX-RAG pipeline is applicable to a broader range of **agentic scenarios**. As discussed in **Appendix.E**, our core mechanism is designed to benefit general-purpose decision-making agents. We explicitly identify that our framework can be extended to more complex agentic tasks, such as **general tool use**, **web navigation**, and **embodied planning**.
> >
> > ---
> >
> > # Response to Q2
> >
> > Please refer to **Response to W1 (q3)**.
> >
> > ---
> >
> > # Response to Q3
> >
> > Please refer to **Response to W3**.

---

> > > ### Comment · Reviewer_4hhY · 2025-11-26
> > >
> > > Thank you for the rebuttal. I will keep my score unchanged.

---

> > > > ### Author Response · Authors · 2025-11-26
> > > >
> > > > Thank you for reading our rebuttal and for the feedback. We are grateful for your constructive comments, and we will ensure that all the concerns raised in your original review are thoroughly addressed and incorporated into our manuscript.

---

### Official Review · Reviewer_DgE2 · 2025-10-30

**Soundness:** 4
**Presentation:** 3
**Contribution:** 4
**Rating:** 4
**Confidence:** 5

**Summary:**

Overall, the paper is interesting and presents a novel idea. It proposes an agentic RL framework that teaches an LLM how to reflect and improve upon its own reasoning. The method leverages a probe policy that, given a trajectory leading to a dead end, generates a new trajectory with reflective reasoning to potentially correct the original mistake. The idea is original, and the overall modeling and writing are clear and well-structured.

However, the main issue is that the authors appear to have omitted the appendix from the submission, which is a serious oversight. This omission makes it difficult to fully understand the paper, particularly regarding several important implementation details.

In summary, I find the paper promising and would be inclined to raise my score to acceptance if the authors can provide the missing appendix and clarify the implementation details during the rebuttal period.

**Strengths:**

- The paper presents an interesting and well-written idea.

- It addresses a compelling question in agentic reinforcement learning: how can we enable LLMs to learn to reflect, or more generally, when introducing an external policy (e.g., for reflection), how can we ensure the policy we want to learn remains on-policy? The paper provides reasonable and well-motivated solutions, including (1) filtering and (2) distribution realignment.

- The experimental evaluation is comprehensive and thoughtfully designed, effectively exploring several important questions related to the framework’s design choices.

**Weaknesses:**

- The appendix pages are missing, which makes it difficult to fully understand several key parts of the work.

- The training process is somewhat complicated and not clearly explained, leading to confusion. For example, it is unclear what the complete training pipeline looks like — whether the two policies are trained jointly or sequentially, and whether they share parameters.

- The ambiguity in describing the training procedure, along with the complexity of the overall training pipeline, makes it difficult to reproduce or fully evaluate the proposed method.

**Questions:**

### Questions for the Authors

- Regarding the definition of *dead ends*: besides trajectories that end with `<answer> ... </answer>`, are there other types of dead-end trajectories considered?

- There is confusion between the *current policy* and the *probe policy*. Does the probe policy $\pi_\epsilon$ share the same parameters as $\pi_\theta$ (only using different prompts), or are they two separate models with distinct parameters?

- For the probe policy, after identifying a dead end, are the subsequent rollouts — including reasoning, search, and answer generation — produced by $\pi_\epsilon$?

- What is the exact training procedure? Are $\pi_\theta$ and $\pi_\epsilon$ trained jointly, or is $\pi_\theta$ first warmed up and then used to train the probe policy?

- Equation (5) defining the probe policy is difficult to interpret. Could the authors provide more explanation for this formulation, particularly why the denominator involves $z^{\frac{1}{|o'_{\text{origin}}|}}$ when $o'_{i,t} \in o'_{\text{origin}}$? What is the intuition behind this design?

- In the ablation study, what exactly is *coarse-PPD*? A one-sentence description is insufficient to understand the difference — please clarify what it looks like in detail.

- It would be helpful to include more **case studies** comparing the behaviors of Search-R1 and the proposed model, to better illustrate their differences.

- During the reflection process, how many new search actions are typically performed?

---

> ### Author Response · Authors · 2025-11-21
> **Response to Reviewer DgE2**
>
> We are sincerely thankful for the encouraging assessment and for recognizing the novelty of our "dead end" solution. The precise and insightful articulation of core motivation, **"ensure the learning policy remains on-policy when introducing an external policy"**, particularly impressed us. We plan to incorporate this exact perspective into revised Introduction to better position our contribution.
>
> We apologize for the confusion regarding the Appendix location. The full Appendix was included **in the Supplementary Material** file, which is allowed by ICLR submission policy (as noted in FAQ of ICLR 2026 Author Guide).
>
> We are pleased to clarify the implementation details below to assist in your assessment.
>
> ---
>
> # Response to W1
>
> As mentioned, the Appendix was placed in the **Supplementary Material**. We will merge these into the main paper in the camera-ready version to ensure better readability.
>
> ---
>
> # Response to W2 & W3
>
> Please refer to **Response to Q2 & Q3 & Q4** below.
>
> ---
>
> # Response to Q1
>
>  In this work, "dead end" is defined as the state where all sampled trajectories fail to reach the correct final answer. We use the Exact Match metric as the trigger signal, which aligns with the Search-R1 baseline. Search-R1 uses Exact Match to trigger its self-reflection mechanism. Therefore, we followed this protocol to **clearly isolate and evaluate our core contribution**.
>
> As discussed in Appendix.E, we acknowledge that this definition is simplistic. It fails to capture flawed reasoning paths that coincidentally yield the correct answer or semantically correct answers that differ in phrasing. We suggest that future work should explore more nuanced triggers, such as **semantic similarity scores** or **uncertainty quantification**, to identify a broader spectrum of reasoning failures.
>
> The development of such granular process supervision is **orthogonal** to our current framework, which focuses on the exploration and policy correction mechanisms triggered by failure. Therefore, it is **compatible with and can be extended** to utilize these signals as triggers in future work.
>
> ---
>
> # Response to Q2 & Q3 & Q4
>
> We apologize for the lack of clarity in our initial manuscript, which caused ambiguity regarding the relationship between the policies.
>
> We clarify that the "Probe Policy" is **not a separate model** with distinct parameters. Instead, it is the **behavioral distribution** when conditioned on an injected Exploration Prompt **under the same model**.
>
> - **Current Policy ($\pi_\theta$)**: Represents the model's standard distribution when generating reasoning trajectories conditioned only on the question and retrieved documents.
> - **Probe Policy ($\pi_\epsilon$)**: Represents the **same model's distribution** when explicitly conditioned on an injected Exploration Prompt. By injecting this prompt into the context window, we shift the generation distribution of the model $\pi_\theta$ to explore alternative paths. Thus, **$\pi_\epsilon$ is functionally an "augmented state" of $\pi_\theta$**.
>
> Since there is only one physical model, there is **no joint training of separate agents**. The training follows a unified iterative loop:
>
> 1. **Sample Collection**: Sample trajectories from the model under standard policy $\pi_\theta$. If a dead end is detected, inject the prompt to sample "probe" trajectories from the same model, which is regarded as pro
> 2. **Optimization**: Then aggregate these trajectories and update the parameters $\theta$. The Policy Correction Mechanism is applied to mathematically correct the distribution shift caused by the prompt injection, ensuring the model learns effectively from this exploration data.
>
> To prevent future confusion, we will revise Section 3.3 in the final paper. We will **explicitly define $\pi_\epsilon$ as a conditional distribution of $\pi_\theta$** given the exploration prompt, and **clearly state that no separate probe network** is instantiated. We will also incorporate the single-loop training process into the main method section to clarify this process.

---

> > ### Author Response · Authors · 2025-11-21
> >
> > # Response to Q5
> >
> > In a word, the term $z^{\frac{1}{|o'_{origin}|}}$ is a renormalization factor designed to **convert the "failure region" of the probability space into a valid, standalone probability distribution**.
> >
> > The "dead end" space represents a subset of the original probability space. The sum of probabilities in this subset is not 1, but $z$, where $z$ means the empirical failure rate and is less than 1. To treat this subset as a valid source distribution for Importance Sampling, we must **"stretch" these probabilities so they sum to 1**. At the trajectory level, this means dividing the probability of the entire sequence by $z$.
> >
> > Since our policy operates at the token level, we cannot simply apply the division once at the end. We must distribute this normalization factor $z$ across the entire sequence of length $| o'\_ {origin} |$. Mathematically, dividing the sequence probability by $z$ is equivalent to dividing each individual token's probability by the $|o'\_{origin}|$-th root of $z$.
> >
> > ### **A concrete example is below.**
> >
> > Suppose a failure trajectory has a length of 2 tokens ($|o'_{origin}| = 2$) and the total failure rate is $z = 0.25$.
> > 1.  We need to increase the trajectory's probability weight by a factor of $\frac{1}{0.25} = 4$ to normalize it.
> > 2.  Then we need to distribute this factor across the 2 tokens. Applying Eq.5, the token-level denominator becomes the square root of $z$: $0.25^{\frac{1}{2}} = 0.5$.
> > 3. For each token, we divide its original probability by 0.5, which is equivalent to multiplying each token by 2. Consequently, across 2 tokens, the total adjustment is $2 \times 2 = 4$, matching the target normalization.
> >
> > This heuristic ensures that the sequence-level renormalization is consistent while preserving the relative probability structure at the token level.
> >
> > ---
> >
> > # Response to Q6
> >
> > Coarse-PPD is an ablation baseline introduced to verify the necessity of precise probability estimation. While our standard method constructs a fine-grained Probability Mass Function based on actual token frequencies, Coarse-PPD **simplifies this by assuming a uniform distribution** over the prompts.
> >
> > To illustrate, consider a prompt pool containing 5 distinct prompts. Coarse-PPD assigns a fixed probability of $1/5$ (20%) to the first token of any chosen prompt and 1 to other tokens, effectively treating all prompts as equally probable regardless of their natural linguistic likelihood.
> >
> > The **2.3% performance drop** observed in Table 2 confirms that this approximation is too rough, and that accurately modeling the probe distribution is critical for the effectiveness of our Importance Sampling correction.

---

> > > ### Author Response · Authors · 2025-11-21
> > >
> > > # Response to Q7
> > >
> > > Thank you the for this valuable suggestion, which can improve the demonstration of our manuscript. We present a detailed comparison below using the question: "Who played the sadistic sheriff in the Clint Eastwood directed film Unforgiven?"
> > >
> > > - Search-R1 gets trapped in a "dead end," performing identical queries repeatedly without strategy adjustment, eventually forcing a wrong answer.
> > >
> > > > Turn 1: `<search>` who played the sadistic sheriff in the Clint Eastwood directed film Unforgiven `</search>` $\rightarrow$ [No direct answer]
> > > >
> > > > Turn 2: `<search>` who played the sadistic sheriff in Unforgiven `</search>` $\rightarrow$ [Same result]
> > > >
> > > > Turn 3: `<search>` who played the sadistic sheriff in Unforgiven `</search>`
> > > >
> > > > Final Answer: "...was played by Richard Johnson." $\rightarrow$ Incorrect
> > >
> > > - REX-RAG recognizes the failure of the specific query and strategically pivots to a broader topic ("cast") before narrowing down again.
> > >
> > > > Turn 1: `<search>` who played the sadistic sheriff... `</search>`
> > > >
> > > > **Reflection**: "...sadistic sheriff... is not directly mentioned. Let's search for more specific information about the cast of the film."
> > > >
> > > > Turn 2: `<search>` cast of Unforgiven `</search>`
> > > >
> > > > **Reflection**: "...cast is mentioned, but not the specific actor... Let's search for more information about the sheriff's actor."
> > > >
> > > > Turn 3: `<search>` sadistic sheriff Unforgiven cast `</search>`
> > > >
> > > > Final Answer: "...found that the actor... is Gene Hackman." $\rightarrow$ [Correct Answer]
> > >
> > > We will include more details in the expanded Appendix.
> > >
> > > ---
> > >
> > > # Response to Q8
> > >
> > > We conducted an **additional analysis tracking the number of valid new searches** performed within unfiltered revision trajectories throughout the training process. We observed three distinct phases in the model's behavior:
> > > - **Phase 1: Initial** (~Steps 0-20) In the early stages, the model struggles to invoke the search engine correctly, resulting in a high failure rate for search calls. Consequently, the count of valid new searches is unstable.
> > > - **Phase 2: Exploration** (~Steps 20-70) As the model learns to utilize the tools, the average number of search actions initially rises to facilitate broader information gathering, and subsequently declines as the policy learns to search more efficiently.
> > > - **Phase 3: Convergence** (~Steps 70-200) After refined its search strategy, the model's behavior stabilizes. In this phase, **approximately 82.5%** of the revision trajectories perform **exactly one** effective new search to correct the reasoning path, while the remainder require multiple searches.
> > >
> > > While the step boundaries vary across different experiments, this overall trend remains consistent. We hope this analysis provides a clearer picture of how the model's search strategy evolves dynamically during the reflection process.
> > >
> > > ---
> > >
> > > We hope these clarifications, along with the details in Supplementary Material, will resolve your concerns. We are fully committed to integrating all the points discussed above into the revised manuscript. We believe these revisions will enhance the clarity, readability, and completeness of our work.

---

### Official Review · Reviewer_X6TS · 2025-11-02

**Soundness:** 2
**Presentation:** 2
**Contribution:** 3
**Rating:** 4
**Confidence:** 4

**Summary:**

This paper investigates the dead-end problem in RAG systems trained with RL, where the model often gets stuck in incorrect reasoning paths and fails to explore new directions. To overcome this, the authors propose REX-RAG, a framework that introduces a Mixed Sampling Strategy to inject exploratory prompts and guide the model toward more diverse reasoning trajectories. However, such exploration may cause distributional shifts that make RL training unstable, so a Policy Correction Mechanism is further designed to re-weight the exploratory data using trajectory filtering and multiple importance sampling, keeping the optimization process stable and unbiased. Experiments on several question-answering benchmarks show that this approach brings consistent performance gains, and ablation results confirm the importance of the correction mechanism for effective exploration.

**Strengths:**

The proposed method seems sound and effective based on the reported results, which makes it convincing that this approach is useful for training policies that better interact with search engines (or tools). Additionally, the experiment setup is very comprehensive and is accompanied by a good set of ablations that clarify the effectiveness of each component in the system.

**Weaknesses:**

The proposed method is significantly more expensive than the baselines, specifically the most similar baseline, search-r1. It is not clear for me the improvements observed here are from the increased number of sampling during training or because of the sampling strategy. We know that number of rollouts in the training can significantly increase the compute budget of training and improving performance. I am curios to  see if this method still performs better than search-r1 if it uses the same number of rollouts (including initial and exploratory rollouts).

**Questions:**

What happens if you assign the same exploration budget (n) to the policy model of search-r1? Would it still downperform your model? Or in other words, I would like to see how your model performs if it can only sample (including exploratory sampling) the same as search-r1? Would that affect your findings?

---

> ### Author Response · Authors · 2025-11-21
> **Response to Reviewer X6TS**
>
> # Response to W1
>
> The central concern is whether the performance gains of REX-RAG primarily come from a larger number of rollouts, rather than from the mixed sampling strategy and policy correction mechanism themselves.
> In the original submission, we already included an analysis in Appendix A.1 (Table 4) that addresses this issue by adjusting the sampling budget. On Qwen2.5-3B, we systematically vary the number of sampled trajectories:
> - **For Search-R1**, we increase the sampling budget by 20% relative to its original configuration, keeping its training procedure unchanged.
> - **For REX-RAG**, we consider both a 12% increase and the default 20% increase in trajectories, using the full exploration and policy correction pipeline.
>
> The results in Appendix A.1 (Table 4) show that **REX-RAG with only 12% more trajectories already outperforms Search-R1 with 20% more trajectories**, and that increasing REX-RAG from 12% to 20% yields further gains, whereas Search-R1 does not see comparable improvement with its 20% increase.
>
> This supports two conclusions:
> - REX-RAG is **more sample-efficient** than Search-R1, with a smaller increase in trajectories leads to larger performance gains.
> - **The performance gains cannot be explained by compute alone.** Simply giving a standard Search-R1 baseline more trajectories does not recover improvements like REX-RAG, indicating that the exploration and policy correction mechanisms are the primary driver.
>
> In the revised manuscript, we will move this critical analysis from the Appendix to the main body to explicitly clarify that the core advantage arises from how trajectories are constructed and used, rather than raw compute.
>
> ---
>
> # Response to Q1
> Please refer to **Response to W1**, and we hope this will address the main concern about disentangling computational budget from algorithmic benefit.

---

### Official Review · Reviewer_iQQ8 · 2025-11-03

**Soundness:** 3
**Presentation:** 2
**Contribution:** 2
**Rating:** 2
**Confidence:** 4

**Summary:**

This paper introduces REX-RAG, a retrieval augmented generation framework to address the dead end problem in RL-based RAG training. That is, when RL rollouts results in incorrect reasoning paths but the policy is unable to self-correct. The proposed REX-RAG consists of (1) a sixed sampling strategy that uses a so-called "probe sampling" to help the model increase rollout sample size to avoid dead ends; and (2) policy correction learning that introduces techniques such as trajectory filtering and multiple importance sampling to stabilize RL training with the introduced rollouts & additional corrections introduced by REX-RAG. The authors evaluate the proposed method on several QA benchmarks, where REX-RAG achieved consistent improvements over RL-based RAG baselines.

**Strengths:**

1. The authors study an important problem of dead end in RL-based RAG settings, where the policy is often unable to generate correct reasoning paths for complex input queries.

2. The authors introduces effective techniques to improve the training dynamics on the rollouts and additional correction continuations by the probe policy.

3. REX-RAG shows strong performance on open-domain QA datasets, suggesting the model learns improved reasoning patterns for multi-turn search LLMs.

**Weaknesses:**

1. The entire exploration and correction mechanism is training-only, as it relies on ground truth labels (rather than learning a verifier) to identify dead ends and trigger exploration. Therefore in inference, these mechanisms are deactivated and the model cannot really correct itself if it heads down to incorrect reasoning paths.

2. The exploration & additional sampling introduces extra computation overhead during the training phase, which may be potentially unfair to baselines like Search-R1 which adopts fixed group size in training. Although the authors provide additional results with over-sampling using DAPO, these results are inconsistent (Search-R1 outperforms REX-RAG) and does have dataset-specific results.

3. Some technical details are missing in writing. For example, although I can image how these are computed, the authors should provide a formula to show how PMF is computed in Eq. 5.

**Questions:**

1. For the exploration prompt sampled from a curated prompt pool, are these tokens masked out in policy update or are they also included in training as in Eq. 5-7?

2. Can you provide more dataset-specific results with DAPO on Search-R1 and REX-RAG?

---

> ### Author Response · Authors · 2025-11-21
> **Response to Reviewer iQQ8:**
>
> We appreciate the time and care you put into assessing our work. However, we believe there are misunderstandings regarding the scope (training vs. inference) and efficiency of REX-RAG, and we would like to clarify this more precisely.
>
> ---
>
> # Response to W1
>
> We confirm that REX-RAG is indeed a training-time framework, but we argue this is a significant strength, not a weakness.
>
> REX-RAG is explicitly proposed as **a training framework, not a test-time self-correction method** (which often suffers from high latency). By leveraging ground-truth labels to guide exploration during training, we allow the model to internalize superior reasoning capabilities, which is analogous to how standard RL utilizes labels/rewards. Consequently, the final model operates using standard decoding without any additional inference cost or reliance on external triggers.
>
> In RL-based RAG, the LLM policy is trained via self-play. When trajectories consistently fall into "dead ends," the policy gradients become uninformative or misleading, leading to policy collapse rather than improvement. Our goal is to make the optimization process robust so that the learned policy itself becomes stronger.
>
> **This interpretation is directly supported by our results.** In Table 1, all reported numbers are obtained **under a standard inference setup**, where all mechanisms are deactivated and there is no access to ground truth. Despite this, REX-RAG achieves consistent gains across all benchmarks (**e.g., +5.1% average gain**), demonstrating that mechanisms transferred reasoning capabilities into the policy weights.
>
> We will clarify in the revised paper that REX-RAG is positioned as a policy optimization method to align with your accurate observation.
>
> ---
>
> # Response to W2
>
> To address your concerns precisely, we have broken down your comment into two specific questions.
>
> ### **(q1) extra computation overhead**
>
> We have already addressed this fairness concern quantitatively in Appendix A.1 (Table 4) in the original submission. To ensure a fair comparison, we conducted a controlled experiment on trajectory budgets:
>
> * **Baseline (Search-R1):** We increased the sampling budget by 20% while keeping the training procedure unchanged. This resulted in negligible performance gains (+0.1%).
> * **REX-RAG:** We evaluated REX-RAG with only a 12% increase in trajectories using our adaptive sampling strategy.
> * **Result:** REX-RAG (+12% samples) significantly outperforms the expanded Baseline (+20% samples) by 2.9% on average (34.2% vs. 31.3%).
>
> This supports the conclusion that REX-RAG is **more sample-efficient** than Search-R1, with a smaller increase in trajectories leading to larger performance gains. In the revised version, we will move this critical analysis to the main paper.
>
> ### **(q2) dataset-specific results on DAPO**
>
> We add per-dataset DAPO results to provide a more granular view in *Table r1*.
>
> | Domain | Dataset | Search-R1 (DAPO) | REX-RAG (DAPO) |
> |:--|:---|:---:|:---:|
> |**General QA**|NQ|46.6|43.4|
> || TriviaQA | 61.9 | 59.3 |
> || PopQA | 44.1 | 42.5 |
> || **Avg.** | **50.9** | **48.4** |
> | **Multi-Hop** | HotpotQA | 32.7 | 36.9 |
> || 2Wiki | 30.4 | 38.8 |
> || Musique | 8.5 | 15.9 |
> || Bamboogle | 19.2 | 32.0 |
> || **Avg.** | **22.7** | **30.9** |
> *Table r1: Detailed comparison on General QA and Multi-Hop QA datasets using DAPO.*
>
> **These dataset-level results align with the results in the main paper on DAPO.** In General QA, REX-RAG is slightly lower than Search-R1. However, in Multi-Hop QA, REX-RAG achieves large gains across all four datasets, leading to an 8.2% improvement on average.
>
> The apparent divergence in results is not an inconsistency but a reflection of an **"Easy-Hard Trade-off"** in the DAPO algorithm. General QA tasks are typically straightforward, requiring single-step retrieval, whereas Multi-Hop QA involves complex reasoning chains where models frequently encounter "dead ends".
>
> Crucially, DAPO is designed to filter out samples with zero advantage, which discards these "dead end" queries. This leads the optimizer to **over-allocate model capacity to the easier General QA** tasks while failing to learn from the harder Multi-Hop problems that it simply ignores.
>
> In contrast, REX-RAG "revives" these dead ends through exploration and policy correction. By generating valid gradient signals for these hard tasks, REX-RAG forces the optimizer to **shift the model's focus and capacity toward solving complex reasoning problems**. This yields massive gains on hard tasks (+8.2%) at the cost of a minor regression on the already saturated easy tasks. Conversely, the GRPO-style methods used in our main results treat all samples uniformly regardless of difficulty, leading to the more symmetrical improvements observed in Table 1.
>
> We will refine the discussion in the main paper to explicitly analyze this trade-off and reference the dataset-specific DAPO table (*Table r1*), offering new insights into sample balance for future RL research.

---

> ### Author Response · Authors · 2025-11-21
>
> # Response to W3
>
> **The complete construction of the PMF is provided in Appendix B.3.** We present the detailed definition of the frequency-based PMF over prompt tokens, and provide the full procedure in Algorithm 1: PMF Construction via Frequency Distribution.
>
> In the revised version, we will add a concise explicit formula for the PMF directly under Eq. 5 and strengthen the pointer to Appendix B.3 and Algorithm 1.
>
> ---
>
> # Response to Q1
>
> **The tokens of the exploration prompts are not masked out.** They are treated as part of the trajectory and included in the policy update, consistent with Eqs. 5-7.
>
> This design allows us to do something that typical test-time scaling methods cannot: we can backpropagate through exploration prompts while maintaining stable and principled RL updates, rather than treating such prompts as purely external hints that must be detached from learning.
>
> ---
>
> # Response to Q2
>
> Please refer to the **q2** of Response to W2.

---

### Author Response · Authors · 2025-12-04
**Summary of Reviews and Rebuttal for the New Area Chair (1/2)**

Dear Area Chair,

We sincerely appreciate you stepping in to handle our submission at this stage. We understand this requires significant effort. To assist in your efficient assessment, we respectfully submit this executive summary. We highlight the consensus on the paper's strengths and clarify **three decisive issues** that heavily influenced the initial ratings.

In this post:
- (1) **Consensus on Strengths:** Reviewers unanimously recognize the novelty and motivation.
- (2) **Three Decisive Issues:** Addressing the decisive factors regarding the Appendix, Efficiency, and Scope.
- (3) **Summary of Remaining Questions:** A comprehensive overview of our responses to all other technical questions.

---

## (1) Consensus on Strengths

Reviewers consistently recognized the significance of the "dead end" problem in RAG and the principled nature of our solution:

* **Novelty & Motivation:**
  * "The idea is original... addresses a compelling question in agentic reinforcement learning." `[DgE2]`
  * "Well-motivated... clearly identifying a key limitation of existing self-reflection methods." `[4hhY]`
* **Empirical Performance:**
  * "REX-RAG shows strong performance on open-domain QA datasets." `[iQQ8]`
  * "Sound and effective... accompanied by a good set of ablations." `[X6TS]`
* **Clarity:**
  * "Well-written... overall modeling and writing are clear." `[DgE2]`

---

## (2) Three Decisive Issues

The initial scores were heavily impacted by the following three points. We have successfully addressed them as follows:

### A. The "Missing" Appendix
`[Reviewer DgE2 (W1)]`

**Issue:**
Reviewer DgE2 rated the paper a 4 primarily because they missed the appendix. They explicitly stated that the omission of the appendix was the main barrier:
> **"I find the paper promising and would be inclined to raise my score to acceptance if the authors can provide the missing appendix."**

**Clarification:**
The Appendix **was included** in the original submission (within the *Supplementary Material* zip file). It contains the derivations (Eq. 5), training pipeline, and prompt details requested. We have pointed the reviewer to this file. **With this misunderstanding resolved, the reviewer's condition for acceptance is fully met.**

### B. Compute Budget
`[Reviewer X6TS (W1/Q1) & Reviewer iQQ8 (W2)]`

**Issue:**
A shared concern of Reviewer iQQ8 and Reviewer X6TS was whether the performance gains derived from an expanded rollout budget; **this was the sole concern for Reviewer X6TS**.

**Clarification:**
We provided a controlled experiment (Table 4) proving **REX-RAG is more sample-efficient**. REX-RAG with only **12% additional samples** significantly **outperforms** the Search-R1 baseline even when the baseline is given **20% additional samples**. This proves the gain comes from the *quality* of exploration, not raw compute budget.

### C. Training vs. Inference
`[Reviewer iQQ8 (W1)]`

**Issue:** Reviewer iQQ8 penalized the method for being "training-only."

**Clarification:** We clarified that this is a **strength**. REX-RAG internalizes reasoning capabilities into the policy, achieving superior performance (+5.1% avg) with **zero additional inference overhead**, unlike expensive inference-time search methods.

---

> ### Author Response · Authors · 2025-12-04
> **Summary of Reviews and Rebuttal for the New Area Chair (2/2)**
>
> ## (3) Summary of Remaining Questions
>
> Beyond the three pivotal issues above, we have addressed all other specific questions raised by the reviewers.
>
> | Reviewer | Issue ID | Question / Concern | Response Summary |
> | :--- | :--- | :--- | :--- |
> | **iQQ8** | **W2/Q2** | Dataset-specific DAPO results | Provided **Table r1**. Explained the "Easy-Hard Trade-off" in DAPO: REX-RAG gains +8.2% on hard Multi-hop tasks by "reviving" dead ends. |
> | **iQQ8** | **W3** | PMF Construction Details | Directed to **Appendix B.3** and **Algorithm 1**, which detail the frequency-based PMF construction. |
> | **iQQ8** | **Q1** | Are prompt tokens masked? | Confirmed they are **not masked**. They are included in training to enable backpropagation through exploration steps. |
> | **DgE2** | **Q1** | Definition of "Dead Ends" | Clarified "Dead Ends" as states where all sampled trajectories fail. |
> | **DgE2** | **W2-W3/Q2-Q4** | Details of Probe Policy | Clarified that Probe Policy is the same model conditioned on prompts, not a separate network. Training uses a unified iterative loop. |
> | **DgE2** | **Q5** | Derivation of Eq. 5 ($z$ term) | Explained that $z$ is a renormalization factor to convert the "failure region" into a valid probability distribution for Importance Sampling. |
> | **DgE2** | **Q6** | Details on Coarse-PPD | Explained it assumes a uniform distribution over prompts; the performance drop proves precise PMF estimation is necessary. |
> | **DgE2** | **Q7** | Case Studies | Added a detailed comparison (e.g., "Clint Eastwood" case) showing REX-RAG pivoting search strategies where baselines fail. |
> | **DgE2** | **Q8** | Count of search actions | Provided analysis of search behavior evolution (Phases 1-3), showing convergence to efficient single-search corrections. |
> | **4hhY** | **W1/W2/Q2** | Self-Reflection Baseline | Clarified that Search-R1 is the SOTA training-time self-reflection baseline (using static triggers). REX-RAG outperforms it by 5.1% via diverse prompt pools. |
> | **4hhY** | **W3/Q3** | Ablation of Mixed Sampling | Explained that removing Mixed Sampling mathematically reverts the model to the Search-R1 baseline. |
> | **4hhY** | **Q1** | Why focus on RAG? | RAG provides a definitive "dead end" (retrieval failure) ideal for validation. We confirmed applicability to broader agentic tasks. |
>
> ---
>
> We deeply appreciate the time and effort you are dedicating to reviewing our case under these unusual circumstances. We genuinely remain confident that this work offers a meaningful contribution to the field and respectfully entrust the final decision to your expert judgment.
>
> Best regards,
>
> Authors of Submission 6827

---

### Note · Authors · 2026-01-26

I have read and agree with the venue's withdrawal policy on behalf of myself and my co-authors.

---

### Meta-Review · Area_Chair_qzAo · 2026-01-03

**Summary:**

The paper addresses an important problem in RL-based RAG, e.g., exploration failure due to reasoning "dead ends", and proposes a framework combining mixed sampling with policy correction. While reviewers agreed on the motivation and potential of the approach, concerns were raised regarding training efficiency, clarity, and the strength of empirical evidence supporting the core claims.

**Reviewer Concerns:**

The rebuttal clarified several points, including the inclusion of the appendix and details of the training pipeline. However, the major concern remains insufficiently resolved: the claim that performance gains are not driven by increased training cost is supported by very limited evidence. During the rebuttal, the authors provide an additional evaluation result to show that REX-RAG with a smaller increase in training samples outperforms Search-R1 given a larger sample budget. However, the evaluation is limited to one model, without showing learning curves, wall-clock analysis, or generalizability across models. Additionally, the paper relies heavily on the appendix for key definitions, limitations, and discussions, making the main text less self-contained.

**Reviewer Scores:**

Reviewer DgE2: (4). Might increase slightly but remains borderline, as efficiency concerns go beyond the appendix clarification.

Reviewer X6TS: Likely unchanged (4), with continued doubt about disentangling rollout count from algorithmic benefit.

Reviewer 4hhY: Unchanged (6), explicitly said would not change score.

Reviewer iQQ8: Likely unchanged (2), maintaining strong concerns about training-only mechanisms and cost comparison despite the authors clarified that no extra inference time mechanisms.

---

### Decision · Program_Chairs · 2026-01-26

Reject